# Landscape heterogeneity buffers biodiversity of simulated meta-food-webs under global change through rescue and drainage effects

Remo Ryser [1,2], Myriam R. Hirt[1,2], Johanna Häussler [1,2], Dominique Gravel[3] & Ulrich Brose [1,2 ✉]

Habitat fragmentation and eutrophication have strong impacts on biodiversity. Meta-community research demonstrated that reduction in landscape connectivity may cause biodiversity loss in fragmented landscapes. Food-web research addressed how eutrophication can cause local biodiversity declines. However, there is very limited understanding of their cumulative impacts as they could amplify or cancel each other. Our simulations of meta-food-webs show that dispersal and trophic processes interact through two complementary mechanisms. First, the 'rescue effect' maintains local biodiversity by rapid recolonization after a local crash in population densities. Second, the 'drainage effect' stabilizes biodiversity by preventing overshooting of population densities on eutrophic patches. In complex food webs on large spatial networks of habitat patches, these effects yield systematically higher bio-diversity in heterogeneous than in homogeneous landscapes. Our meta-food-web approach reveals a strong interaction between habitat fragmentation and eutrophication and provides a mechanistic explanation of how landscape heterogeneity promotes biodiversity.

[1] German Centre for Integrative Biodiversity Research (iDiv) Halle-Jena-Leipzig, Leipzig, Germany. [2] Institute of Biodiversity, Friedrich Schiller University Jena, Jena, Germany. [3] Département de Biologie, Université de Sherbrooke, Sherbrooke, QC, Canada. ✉email: ulrich.brose@idiv.de

Increasing human demands for production of goods in natural landscapes have caused habitat fragmentation and homogenisation, eutrophication, and increasing land-use intensity. This resulted in an erosion of biodiversity and associated ecosystem services at global scales. Habitat fragmentation dissects continuous natural landscapes into habitat patches embedded in a matrix whose hostility (i.e. dispersal mortality) for the species increases with land-use intensity. Increasing nutrient inputs from agricultural practices yield biomass accumulations at higher trophic levels, eroding biodiversity by increased species' interaction strengths[1,2]. Despite growing evidence of these global change factors' importance, we still do not understand how their interaction drives biodiversity changes. While fragmentation and eutrophication are often studied in isolation, complex feedback loops in multi-trophic food webs can generate non-linear responses in biodiversity. This is rendering our knowledge of the interactive effects of these stressors in natural landscapes fraught with uncertainty. The high-dimensional interplay between spatial and trophic processes prevents experimental studies on such complex interactions. Therefore, simulations of spatial food-web dynamics are needed to reveal the mechanisms underlying how these global change stressors interact.

One key challenge is integrating spatial processes connecting local communities across habitat patches into metacommunities and interaction processes connecting species into complex food webs (Fig. 1). Traditionally, independent and mostly separated research areas have addressed these two types of ecological networks. First, metacommunity theory[3,4] describes how dispersing individuals connect local communities across complex spatial networks of habitat patches[5]. Depending on their size and quality, patches can comprise large source populations that yield a net dispersal flux of individuals to small sink populations[6–8] (Fig. 1a). These source-sink dynamics[9] can facilitate the persistence of small populations by rescue effects[10], which is undermined by increasing fragmentation or land-use intensity that prevent successful dispersal. Second, food-web theory addresses how biomass fluxes (i.e. energy and matter) between species drive community dynamics[11] (Fig. 1b). Weak biomass fluxes, relative to the populations' loss rates, can cause consumer extinction due to energy limitations. In contrast, strong biomass fluxes relative to loss rates can result in top-heavy consumer-resource biomass pyramids with unstable dynamics[1,2]. Eutrophication, in particular, increases all biomass fluxes yielding top-heavy food webs and thus undermines the biodiversity of local communities[12]. Although both research areas documented the strongly negative effects of either fragmentation or eutrophication on biodiversity, the interplay of these stressors in complex natural communities has remained virtually untapped.

So far, studies synthesising spatial and trophic processes have been limited to small modules such as food chains[13,14]. These studies showed that dispersal can synchronise population dynamics across habitat patches, which reduces biodiversity by correlated local extinctions[15,16]. However, dispersal of consumers can also prevent extinctions by inducing compensatory

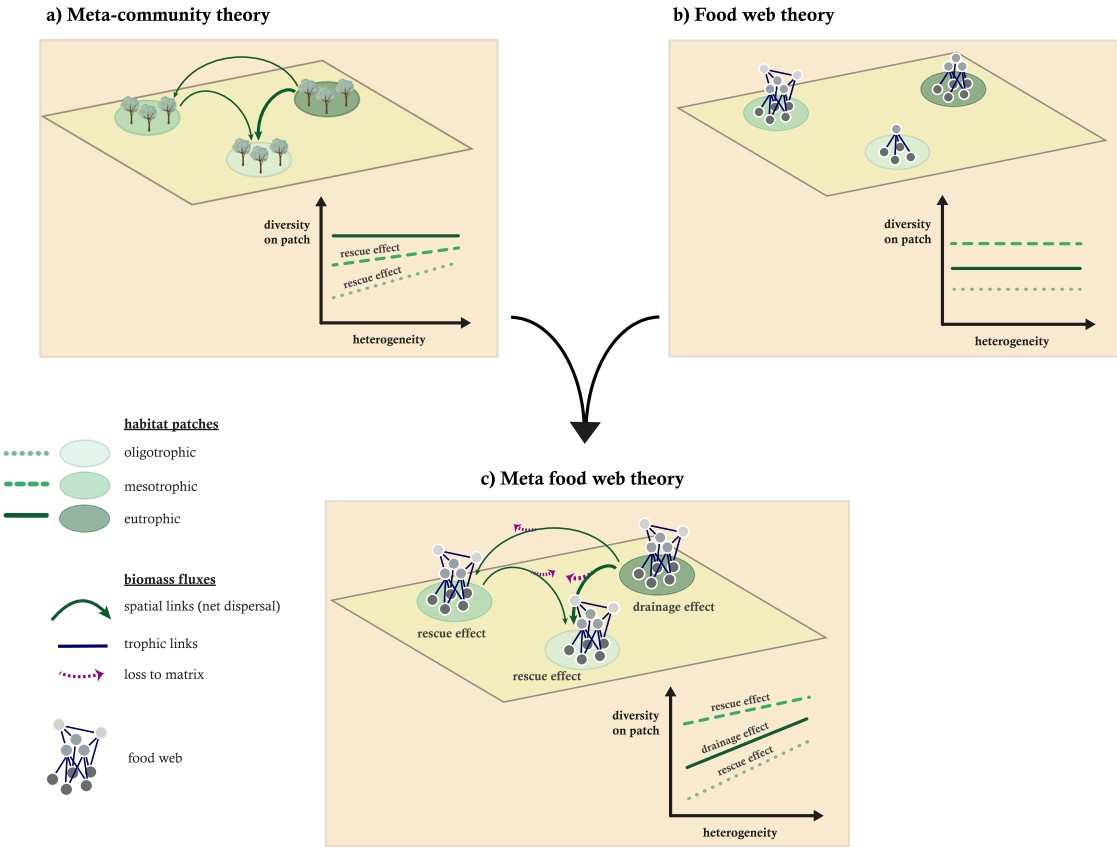

**Fig. 1 Conceptual figure illustrating the synthesis of metacommunity theory and food-web theory into meta-food-web theory.** Panel **a** illustrates metacommunity dynamics with net dispersal from larger (nutrient richer patches; darker green) to smaller populations (nutrient poorer patches; lighter green) and the associated rescue effect on local diversity. Panel **b** illustrates local food-web dynamics on patches with different nutrient richness (shades of green) and the effect of the paradox of enrichment on local diversity. Panel **c** illustrates the synthesis of metacommunity and food-web dynamics and the interaction of rescue and drainage effects and their consequence for biodiversity. Heterogeneity arises as patches with different nutrient supplies occur, which yields **a** rescue effects on oligotrophic and mesotrophic patches in metacommunities, **b** patterns with the highest biodiversity on mesotrophic patches in food webs, and **c** rescue (oligotrophic patches) and drainage effects (eutrophic patches) in meta-food-webs.

dynamics[17], indirect negative density dependence that dampens oscillations[18–20], and attenuation of strong interspecific interactions[21]. Even in these simple interaction modules, the relative strength of these positive and negative effects of dispersal on population persistence depends on the trophic level that is dispersing[18] and the trophic interaction structure[14]. These findings render the analysis of dispersal impacts on biodiversity in large spatial networks with many species essential.

Traits of organisms play an essential role in both spatial and trophic processes. In metacommunities, body mass and movement mode determine which patches compose species-specific spatial networks[22]. Similarly, the propagation of energy fluxes through food webs is driven by species' interaction strengths that depend strongly on body masses[23]. Although metacommunity and food-web theories have been developed mostly independently, they have identified the same important drivers (i.e. body mass) and the same currencies (i.e. biomass fluxes). To date, a trophic metacommunity framework incorporating spatial use properties is still lacking[24]. Also, as spatial and trophic processes in natural landscapes are coupled (Fig. 1c), a mechanistic understanding of global change effects on ecosystems will benefit from an integrated approach.

We address this challenge by synthesising metacommunity and food-web models. Using a bioenergetic model, we analyse population dynamics across a gradient of complexity from simple (tri-trophic food chain on a single patch) to complex systems (40-species food web on 50 habitat patches). This model employs a single and easy to measure trait, body mass, as the unifying characteristic that determines not only trophic links and interaction strengths of the food webs but also the dispersal ranges. Dispersal rates depend on local net growth rates, summarising resource availability, competition, and predator pressure arising from local trophic dynamics[25]. By predominantly relying on empirically derived body mass scaling relationships of processes, this model remains relatively parsimonious despite its complexity. Analyses of this meta-food-web model identify a key mechanism, referred to as the 'drainage effect', that complements the rescue effect in landscapes under eutrophication and fragmentation (Fig. 1). We show that biodiversity is safeguarded by well-known rescue effects creating positive net migration fluxes (i.e. more biomass is immigrating than emigrating) into oligotrophic patches and drainage effects implying negative net migration fluxes (i.e. more biomass is emigrating than immigrating) out of eutrophic patches (Fig. 1).

## Results and discussion

**Drainage effect on a single patch.** First, on a single patch, low nutrient supply for a tri-trophic food chain causes predator starvation (Fig. 2a, extinction, left side). Increasing nutrient supply first promotes predator equilibrium biomass densities (Fig. 2a, survival, equilibrium) and therefore top-heavy biomass pyramids causing biomass oscillations (Fig. 2a, survival, oscillation), which paradoxically eventually yield predator extinction (Fig. 2a, extinction, right side). Such extinctions due to unstable oscillations under eutrophication have first been described as the paradox of enrichment[1]. Subsequently, they were generalised to systems with an increased energy flux to the predator relative to its loss rate[2,26]. Turning around this 'principle of energy flux' then suggests that an additional drainage effect arises from spatial energy loss. In this vein, theory highlighted the potential of consumer dispersal as generally strong stabilising effects of direct density-dependent negative feedbacks to prevent extinctions[18,21,27]. Consistent with this, we find that increasing emigration rates that drain biomass out of a eutrophic location can prevent predator extinction by reducing oscillations (Fig. 2b). In this single patch

scenario, the drainage effect only comprises of dispersal loss (as all emigrating biomass is lost to the matrix) and is mathematically equivalent to an additional source of mortality. This effect also holds across different dispersal models but is amplified by active dispersal (e.g. adaptive and non-adaptive dispersal; see the Supplementary Notes for sensitivity analyses). Spatial fluxes increase with dispersal rates and the underlying variability in the landscape. This demonstrates the drainage effect as a mechanism by which spatial processes can stabilise trophic population dynamics in heterogeneous landscapes.

**The drainage effect with two patches.** Subsequently, we studied this drainage effect in systems of two connected habitats across gradients of landscape hostility (i.e. dispersal loss) and habitat heterogeneity (represented by the difference in nutrient supply concentration between the two locations). Landscape hostility summarises all factors that drive the loss of biomass during dispersal including energetic costs of movement and dispersal mortality (e.g. road kills) in the unsuitable landscape matrix. In simulations without heterogeneity (two eutrophic patches) and without dispersal loss, dispersal synchronises unstable dynamics causing predator extinction (Fig. 3, lower left corner).

When two patches of different quality are connected by dispersal, the drainage effect araises from two different processes. (1) dispersal loss as described above, and (2) from spatial energy transfer from large populations (sources) to small populations (sinks), essentially transfereing parts of the mortality rate of the sink population indirectly, and preventing unstable dynamics in top-heavy systems. Thus, increasing landscape hostility (i.e. dispersal loss) yields drainage of biomass during dispersal, which first facilitates predator persistence and then also reduces oscillations (Fig. 3, along the dispersal loss axis). At very high levels of landscape hostility, however, extreme death rates during dispersal cause predator extinction. Similarly, increasing patch heterogeneity (reducing the nutrient suppy on one patch) also enables predator persistence and decreases oscillations (Fig. 3, along the heterogeneity axis). At the level of local habitat patches, these findings generalise the mechanisms of the drainage effect (Fig. 2) on local biodiversity across variation in dispersal loss and patch heterogeneity (Fig. 3; for results on patterns in population oscillations see Supplementary Fig. 2). For eutrophic patches, increased dispersal losses by landscape hostility or the coupling with an oligotrophic patch (patch heterogeneity) both increase the biomass drainage through increased net migration (i.e. more emigration than immigration). For oligotrophic patches, however, there are differences between effects of landscape hostility and patch heterogeneity. Drainage by landscape hostility (i.e. dispersal loss) supresses small populations even more, whereas patch heterogeneity causes a gain in biomass via dispersal that supports predator populations via rescue effects and trophic cascades[28] (see Supplementary Fig. 1). Patch heterogeneity thus creates dispersal fluxes in biomass that are responsible for not only the well-known rescue effects[10] supporting small populations on oligotrophic sink patches by net-immigration but also the drainage effects sustaining large populations on eutrophic patches.

**Complex food webs and complex landscapes.** To generalise the mechanistic understanding of drainage effects from food chains, we simulated the dynamics of a complex food web consisting of 10 plants and 30 animals on different complex landscapes containing 50 habitat patches (Fig. 4). We simulated homogeneous landscapes, where all patches have the same nutrient supply concentration. These simulations were replicated across a gradient of nutrient supply concentrations ranging from $10^{-0.8}$

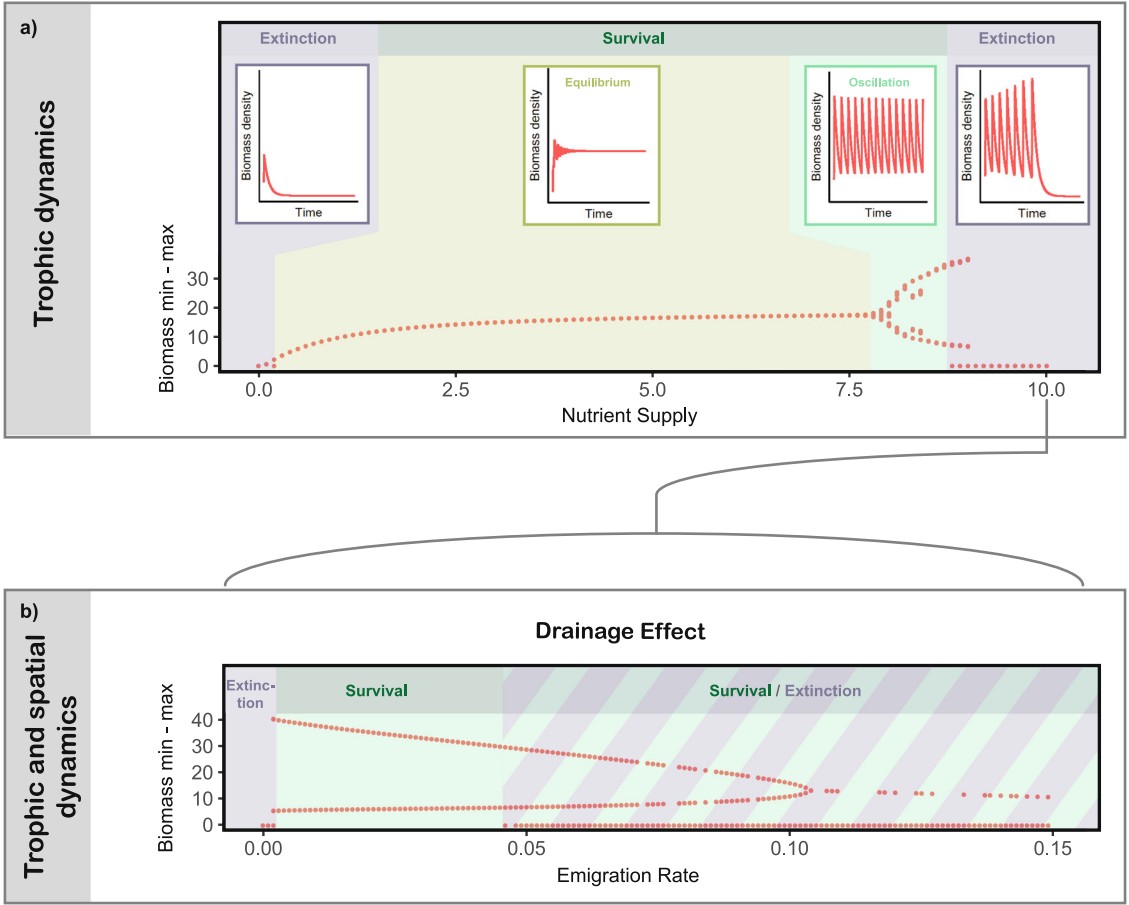

**Fig. 2 Top predator dynamics of a tri-tropic food chain on a single patch. a** Bifurcation diagram and exemplary time series of biomass densities of the predator at different nutrient supply concentrations (boxes; from left to right: 0.1 (oligotrophic); 3 (mesotrophic); 8.5 and 10 (eutrophic)) corresponding to points in the bifurcation diagram showing maximum and minimum biomass density (y-axis) across a gradient of nutrient supply concentrations (x-axis). **b** Bifurcation diagram showing maximum and minimum biomass density (y-axis) when enabling emigration across a gradient of maximum emigration rates (x-axis; ***a*** in Eq. (10) in the Methods) with a nutrient supply concentration of 10, which corresponds to the last point in panel **a**.

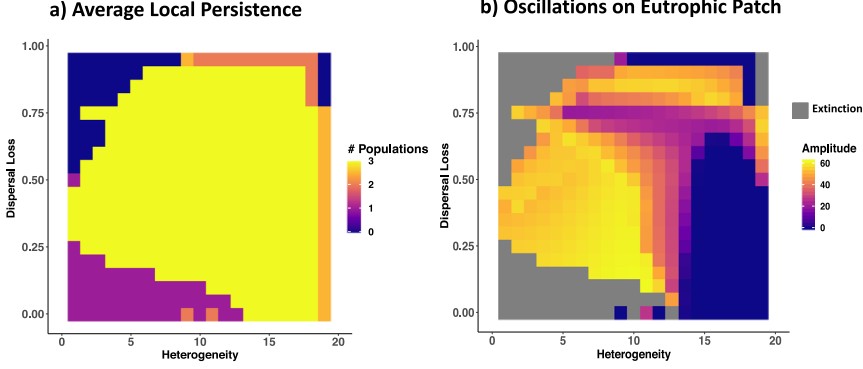

**Fig. 3 Top predator dynamics of a tri-tropic food chain on two coupled patches. a** Heatmap showing the average number of persisting populations (colour coded; plant, herbivore and predator; maximum of 3) of the two patches across gradients of landscape heterogeneity (x-axis; difference in nutrient supply concentration across the two patches; on the left: two eutrophic patches, on the right: a eutrophic and an oligotrophic patch) and dispersal loss (y-axis). **b** Heatmap showing the amplitude of biomass density oscillations of the predator (z-axis; colour coded) in the (always) eutrophic patch across gradients of landscape heterogeneity (x-axis; difference in nutrient supply concentration between the two patches) and dispersal loss (y-axis). Amplitudes of 0 (blue) stand for an equilibrium state of the predator. Grey areas are where the predator went extinct.

(oligotrophic) to $10^2$ (eutrophic). We also simulated three types of heterogeneous landscapes with landscape averages being oligotrophic, mesotrophic, or eutrophic (Fig. 4). Nutrient supply concentration for each patch of heterogenous landscapes is assigned randomly from the same gradient as the range of nutrient

supply concentrations in the homogeneous scenarios ($10^{-0.8}$–$10^2$), with a higher sampling probability in the lower or higher nutrient supply values for oligotrophic and eutrophic heterogeneous landscapes, respectively, and uniform sampling for the mesotrophic heterogeneous landscapes. In line with our results from the food-

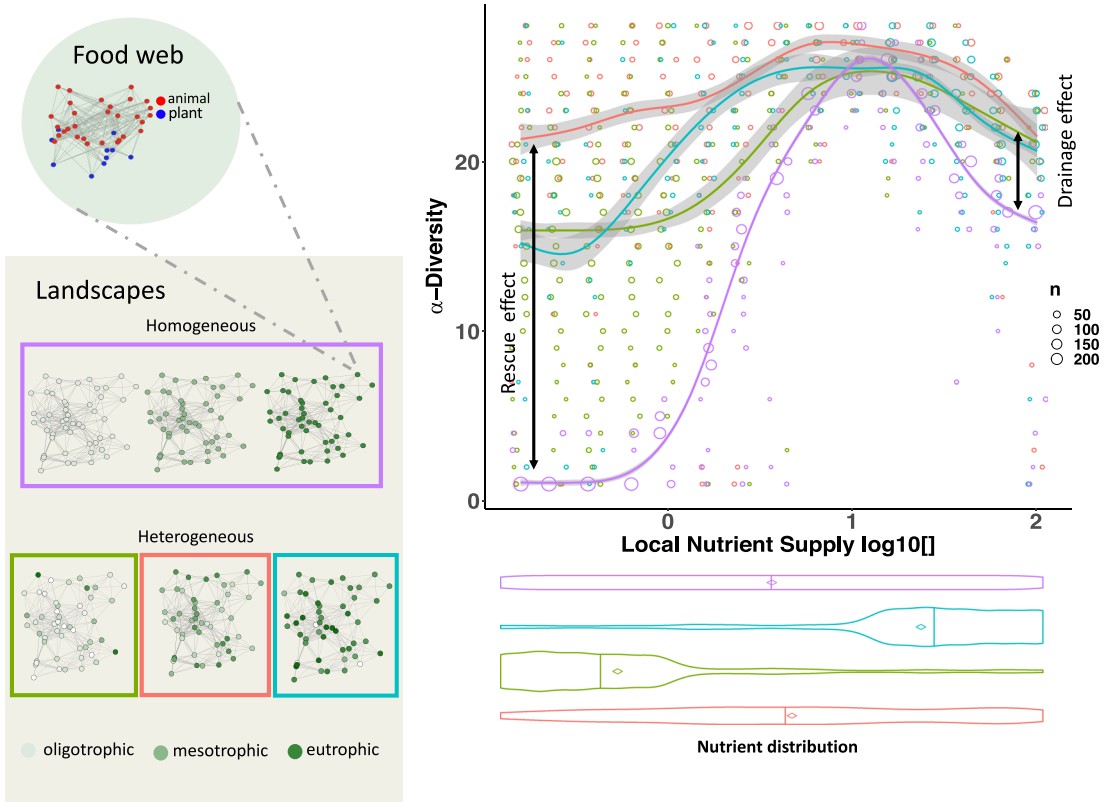

**Fig. 4 Landscape heterogeneity drives biodiversity in complex meta-food-webs.** Local diversity on a patch (*y*-axis) across a gradient of local patch nutrient supply concentration (nodes in shades of green) in homogeneous (purple) and heterogeneous (green, orange, blue) landscapes. Violin plots below the *x*-axis show nutrient distributions within the landscape for each scenario, bars represent medians and diamonds represent means. The meta-food-web consists of a complex food web of 10 plants (blue nodes) and 30 animals (red nodes) and large homogeneous and heterogeneous landscapes with 50 habitat patches with different patch nutrient supply concentrations (nutrient supply concentrations on habitat patches are colour coded). Edges indicate dispersal links for an exemplary species with a dispersal range of 0.3. Lines are a smooth fit from a GAM model with 95% confidence intervals in ggplot2, circles represent the data and the circle size the number of data points.

chain simulations, we find that local species richness in homogeneous landscapes is lowest on oligotrophic patches due to energy limitation. Higher nutrient supply first increases species richness on mesotrophic patches before decreasing it again on eutrophic patches (Fig. 4, purple). Local species richness is highest on habitat patches in mesotrophic heterogeneous landscapes because oligotrophic patches profit from the rescue effect, and eutrophic patches profit from the drainage effect (Fig. 4, orange). If there are only a few oligotrophic patches in a eutrophic heterogeneous landscape, rescue and drainage effects still increase local diversity, although the rescue effect is weaker (Fig. 4, blue). Similarly, a few eutrophic patches in an oligotrophic landscape foster local diversity through rescue and drainage effects (Fig. 4, green). Note that we present the consequences of rescue and drainage effects for alpha diversity of local patches, whereas effects on gamma diversity of the whole landscape remain to be addressed. Nevertheless, our results demonstrate that rescue and drainage effects also apply to complex food webs in complex landscapes. This shows that the interaction of strong and weak spatial and trophic biomass fluxes increases stability and species richness in metacommunities.

**Implications and perspectives**. Spatial processes in heterogenous landscapes stabilise local food-web dynamics and translate into higher alpha diversity on the habitat patches. This stresses the importance of addressing global change drivers in a meta-food-web framework. Various mechanisms are involved, all related to source-sink dynamics, where individuals move from high biomass locations to low biomass locations. We have found that the

well-known rescue effect allows persistence on oligotrophic patches, while the drainage effect buffers eutrophic patches. Complex interactions among these phenomena may further promote diversity at a local (e.g. patch) and a regional (e.g. landscape) scale. In this study, we have focused on the local scale to obtain a mechanistic understanding of drainage and rescue effects. Thereby, we gain insight into how landscape conditions change dispersal fluxes depending on local nutrient conditions and trophic energy fluxes.

While our study has been focused on the biodiversity of local patches, some of the results have implications for biodiversity at the landscape level. We found consistently higher local diversity in mesotrophic heterogeneous landscapes (Fig. 4, orange line) than in homogeneous landscapes with the same landscape average of nutrient supply (the local diversity on the purple line at the average nutrient supply level of orange landscapes indicated by the orange diamond in the violin plot). This finding suggests positive synergies between drainage and rescue effects. Across these simulations heterogeneity in nutrient supply of the patches is mostly correlated with changing the landscape-level average nutrient supply (but see Supplementary Fig. 7 for landscape-average nutrient supply independent heterogeneity). Increasing or decreasing the landscape-average nutrient supply does have implications on total energy input and thus on stability. Therefore, unbiased comparisons across landscapes should thus be restricted to the same landscape average of nutrient supply (diamonds in the violin plots relative to purple line of Fig. 4). Additionally, further comparisons of regional diversity between

homogeneous and heterogeneous landscapes also need to consider the spatially explicit locations and distances of oligotrophic and eutrophic patches in the spatial networks. Furthermore, nutrient spillover from a eutrophic to a neighboring oligotrophic location may promote local productivity and increase food-chain length[29]. Such spatial nutrient diffusion can destabilise simple food chains and decrease spatial heterogeneity in a meta-ecosystem model[18], and thus cross-ecosystem nutrient fluxes can change community composition[30]. These meta-ecosystem approaches have synthesised nutrient fluxes with simple trophic modules. Our meta-food-web approach provides a flexible tool to scale-up these processes to the levels of landscape and food-web complexity that characterise natural ecosystems.

In real landscapes that suffer increasingly from fragmentation, land-use intensification, and eutrophication due to human activities, managing connectivity and heterogeneity is an essential aspect of biodiversity conservation and restoration. Traditionally, increasing landscape hostility due to higher dispersal mortality or increased distances between habitat fragments have been perceived as threats to the biodiversity of habitat patches as they reduce rescue effects[10]. Hence, wildlife bridges across highways and other corridors that increase connectivity between habitat patches have been propagated as important tools to remedy the consequences of land-use intensification. The reduced hostility may benefit small sink populations by rescue effects and thus lower extinction risks[31]. Our results, however, indicate that the consequences of increasing habitat connectivity are highly context dependent. We found that higher connectivity between large populations can undermine biodiversity by decreasing the drainage effect, whereas connecting large and small populations is generally beneficial for both. Thus, in managing landscape connectivity, connections between eutrophic and oligotrophic habitats or among oligotrophic habitats should be enhanced to reduce the hostility effect. However, connections among eutrophic habitats should only be established with caution, as a reduced hostility effect results in less drainage effect and can destabilise both populations.

While dispersal generally invokes an indirect negative density dependence, it is a complex process, and many more ecological relevant traits and processes may affect the magnitude of dispersal fluxes[24]. For instance, we also anticipate that future meta-food-web models could include traits beyond body mass such as movement mode (e.g. flying, running and swimming)[32], predation strategies (e.g. group hunting), or cognitive ability[24] to describe dispersal more realistically. While we performed simulations with different dispersal models that did not change the general patterns caused by drainage and rescue effects (see Supplementary Figs. 3–6), all of these dispersal models assume that the dispersing organisms have no information about their destination. However, information about conditions on dispersal destinations may affect decisions to and where to disperse. We anticipate that this could further stabilise meta-food-webs as dispersal biomass fluxes would be directed to the habitat patches with the best conditions. Additionally, connectivity between habitat patches may also yield fluxes other than energy and biomass, such as traits or genes. A possible trade-off between the benefits of reduced energy fluxes for eutrophic patches and the negative consequences of reduced gene flow remains to be investigated. In general, the effects emerging in our study call for empirical validation. Our modeling results provide testable predictions, most realistically to be realized in microcosm and mesocosm studies where local densities and dispersal can be monitored, on how enhanced dispersal can stabilise or destabilise local communities. However, as long as field experiments at larger scales are logistically impossible our meta-food-web model offers mechanistic understanding how spatial and trophic processes interact in driving biodiversity patterns.

Broader implications for ecosystem services can arise as two habitat patches that suffer from eutrophication may lose predatory pest control agents if they are well connected but may maintain pest control when coupled with less intensive or natural habitats. Thus, the management of connectivity and heterogeneity in landscapes suffering from fragmentation and eutrophication may foster rescue and drainage effects to maintain biodiversity and ecosystem services. Our meta-food-web approach has revealed interactions between spatial and trophic dynamics beyond the well-known rescue effect. Our results provide a mechanistic explanation of how landscape heterogeneity enhances biodiversity, facilitating new strategies for active landscape management to foster natural biodiversity and ecosystem services.

## Methods

**Model**. We model a tri-trophic food chain of one plant, one herbivore and one predator population on one or two habitat patches and complex meta-food-webs consisting of 10 plants and 30 animals in different landscapes containing 50 patches. The feeding dynamics are constant overall patches and are determined by the allometric food-web model by Schneider et al. 2016[33]. We integrate dispersal as species-specific biomass flux between habitat patches according to Ryser et al. 2019[34]. With the use of a dynamic bioenergetic model we formulate feeding and dispersal dynamics in terms of ordinary differential equations. The rate of change in biomass densities of a species are the sum of its biomass loss by metabolism, being preyed upon and emigration and its biomass gain by feeding and immigration. For detailed equations and for model parameters see section Equations and parameters and the supplement (Supplementary Table 1).

**Local food-web dynamics**. Following the allometric food-web model by Schneider et al. 2016[33] each species is fully characterised by its average adult body mass. For the complex food-web $\log_{10}$ body masses were randomly drawn from a uniform distribution from 0 to 3 for plants and from 2 to 6 for animals. For the food chain the plant body mass was set to $10^2$, the herbivore body mass to $10^4$ and the predator body mass to $10^6$. We set mass ratios of the herbivore to the plant and the predator to the herbivore to the optimum of 100, thus the respective resource being a one-hundredth of its consumer's body mass. This simplifies feeding efficiency rates (see section Equations and parameters; $L_{i,j}$, Eq. (5)) to 1 in the case of a food chain. Trophic dynamical parameters, such as metabolic rates and feeding rates, scale with body masses of model species. Also, we assume a type-II functional response for the food chain and a slight nonlinearity of the functional response in the food web as this stabilises persistence in more complex systems. Compared to Ryser et al 2019, capture rates were reduced to 5% to achieve viable food chains and food webs to increase the stability in the absence of interference competition.

**Nutrient model**. We have an underlying nutrient model with one nutrient that is driving the nutrient uptake and therefore the growth rate of the plant population[11,33]. The nutrient model consists of one nutrient, a nutrient turnover rate of 0.25 and a nutrient supply concentration. The nutrient supply concentration was varied to get eutrophic and oligotrophic patches (see Setup).

**Spatial dynamics**. We model dispersal between local communities as a dynamic process of emigration and immigration, assuming dispersal to occur at the same timescale as the local population dynamics[35]. Thus, biomass flows change dynamically between local populations and the dispersal dynamics directly influence local population dynamics and vice versa[25].

Dispersal rates of animals are modelled with an adaptive emigration rate depending on the net growth rate on the given patch. Dispersal ranges depend on the body masses of our model species with larger species having a higher dispersal range. We model a hostile matrix between habitat patches that does not allow feeding interactions to occur during dispersal. Depending on the scenario, we define a landscape with one, two or 50 patches. In cases with two or 50 patches, their locations are spatially explicit and were chosen in a way that the distances between reflect the dispersal loss of the predator across the matrix hostility gradient.

**Emigration and immigration**. Based on empirical observations[36] and previous theoretical frameworks[13,22,37], we assume that the maximum dispersal distance of animal species increases with their body mass. For simplicity, we do not let the plants disperse, as they do not move themselves and the dispersal of plant propagules strongly depends on their dispersal strategy. We model emigration rates as a function of each species' per capita net growth rate, which is summarising local conditions such as resource availability, predation pressure, and inter- and intra-specific competition[25] (but see Sensitivity Analyses for dispersal models with constant dispersal or non-body-mass-scaled dispersal ranges). Dispersal losses scale

linearly with the distance between two patches and are 100% in scenarios with only one patch or when the distance between the two patches surpasses the dispersal range of an animal. Even though we model dispersal losses according to dispersal distances, this loss term could also represent any other sort of dispersal loss. For numerical reasons, we did not allow dispersal flows smaller than $10^{-10}$.

**Numerical simulations**. We initialised each local population with a biomass density randomly sampled from a uniform probability density within the interval (0,10). Starting from these random initial conditions, we numerically simulated food web and dispersal dynamics over 100,000 time steps by integrating the system of differential equations implemented in C++ using procedures of the SUNDIALS CVODE solver version 2.7.0 (backward differentiation formula with absolute and relative error tolerances of $10^{-10}$) and the time series of biomass densities were saved for last 10,000 time steps. For numerical reasons, a local population was considered extinct and was set to 0 once its biomass density dropped below $10^{-20}$. Based on the empirically derived metabolic rates, these 100,000 time steps correspond to ~11 years. Our model does, however, not account for time spent for organisms' other non-trophic activities such as sleeping or mating. Thus, the time scales of the simulation should only be compared with caution to natural time scales of population dynamics. Transient dynamics usually equilibrate within the first few thousand time steps.

**Equations and parameters**. Our model formulates the change of biomass densities over time in ordinary differential equations. Given the empirical origin of metabolic rates used in our model, one time step corresponds to an hour and body masses are in mg, areas of patches are not defined. The feeding links (i.e. who eats whom) are constant overall patches and are as well as the feeding dynamics determined by the allometric food-web model by Schneider et al. 2016[33]. We integrate dispersal as species-specific biomass flow between habitat patches. Using ordinary differential equations to describe the feeding and dispersal dynamics, the rate of change in biomass density $B_{i,z}$ of species $i$ on patch $z$ is given by

$$\frac{dB_{i,z}}{dt}[mg * \text{Area}^{-1} * h^{-1}] = B_{i,z}\sum_j e_j F_{ij,z} - \sum_j B_{j,z}F_{ji,z} - x_i B_{i,z} - E_{i,z} + I_{i,z}(\text{for animals})$$

(1)

$$\frac{dB_{i,z}}{dt}[mg * \text{Area}^{-1} * h^{-1}] = r_i G_i B_{i,z} - \sum_j B_{j,z}F_{ji,z} - x_i B_{i,z}(\text{for plants})$$

(2)

with the first three terms describing local trophic dynamics and the last two terms describing emigration, $E_{i,z}$ (Eq. 9), and immigration, $I_{i,z}$ (Eq. 11). For simplicity, we do not let plants disperse. Trophic dynamics are driven by following three processes. First, predation or herbivory on species $j$ with assimilation efficiency $e$ ($e_j = 0.545$, if $j$ is a plant, typical for herbivory; $e_j = 0.906$ if $j$ is an animal, typical for carnivory[38]) and the functional response $F_{ij,z}$ (Eq. 3) for animals, and a nutrient dependent growth (Eq. 7) for plants. Second, losses due to predation or herbivory, respectively. Third, losses by metabolic demands with $x_i = x_A m_i^{-0.305}$ with scaling constant $x_A = 0.141$ (tenfold laboratory metabolic rate[39] at a temperature of 20° Celsius to represent field metabolic rates) for animals and $x_i = x_P m_i^{-0.25}$ with $x_P = 0.138$ for plants. We used a dynamic nutrient model (Eq. 8) as the energetic basis of our food web. Each species $i$ is fully characterised by its average adult body mass $m_i$. Body masses determine the interaction strengths of feeding links as well as the metabolic demands of species. Data from empirical feeding interactions are used to parametrise the functions that characterise the optimal prey body mass and the location and width of the feeding niche of a predator[33]. From each $m_i$ a unimodal attack kernel, called feeding efficiency $L_{ij}$ is constructed which determines the probability of consumer species $i$ to attack and capture an encountered resource species $j$. We model $L_{ij}$ as an asymmetrical hump-shaped Ricker's function (Eq. 5) that is maximised for an energetically optimal resource body mass (optimal consumer-resource body mass ratio $R_{opt} = 100$) and has a width of $\gamma$. The maximum of the feeding efficiency $L_{ij}$ equals 1. Supplementary table 1 is an overview of the standard parameter set for the equations. See also Schneider et al. 2016[33] for further information regarding the allometric food-web model.

Functional response

$$F_{ij,z} = \frac{\omega_i b_{i,j} R_{j,z}^{1+q}}{1 + \omega_i \sum_k b_{ik} h_{ik} R_{k,z}^{1+q}} \cdot \frac{1}{m_i}$$

(3)

Per unit biomass feeding rate of consumer $i$ as function of the biomass density of the resource $R_j$, with $b_{i,j}$, resource-specific capture coefficient (Eq. 4); $h_{i,j}$, resource-specific handling time (Eq. 6); $\omega_i = 1/(\text{number of resource species of } i)$, an inefficiency parameter for generalists assuming that generalist are less adapted in for example search patterns or hunting strategies to a specific prey species; and $q$, the Hill coefficient for nonlinearities in density dependency (if $q = 0$ it is a Type-II functional response, if $q = 1$ it is a Type-III functional response).

Capture coefficient

$$b_{ij} = f a_k m_i^{\beta_i} m_j^{\beta_j} L_{ij}$$

(4)

Resource-specific capture coefficient of consumer species $i$ on resource species $j$ scaling the feeding kernel $L_{ij}$ by a power function of consumer and resource body mass, assuming that the encounter rate between consumer and resource scales with

their respective movement speed. This body mass scaling of encounter rates is assumed to occur before the attempt of a predator to capture its prey is made. We differentiate between carnivorous and herbivorous interactions with each comprising a constant scaling factor for their capture coefficients $a_k$ with $k \in 0, 1$ ($a_0 = 15$ for carnivorous species and $a_1 = 3500$ for herbivorous species). For plant resources, $m_j^{\beta_j}$ was replaced with the constant value of 1 (as plants do not move).

Feeding efficiency

$$L_{i,j} = \left(\frac{m_i}{m_j R_{opt}}e^{1-\frac{m_i}{m_j R_{opt}}}\right)^\gamma$$

(5)

The probability of consumer $i$ to attack and capture an encountered resource $j$ (which can be either plant or animal), described by an asymmetrical hump-shaped curve (Ricker's function), centered around an optimal consumer-resource body mass ratio $R_{opt} = 100$[33] and with $\gamma$ that that affects the width of the hump. An increase in $\gamma$ results in a decrease in the width.

Handling time

$$h_{ij} = h_0 m_i^{\eta_i} m_j^{\eta_j}$$

(6)

The time consumer $i$ needs to kill, ingest, and digest resource species $j$, with scaling constant $h_0 = 0.4$ and allometric exponents $\eta_i = -0.48$ and $\eta_j = -0.66$[40].

Growth factor for plants

$$G_i = \frac{N}{K_i + N}$$

(7)

Species-specific growth factor of plants determined dynamically by the nutrient; with $K_i$, half-saturation densities determining the nutrient uptake efficiency assigned randomly for each plant species i and (uniform distribution within (0.1, 0.2)).

Nutrient dynamics

$$\frac{dN_z}{dt} = D(S - N) - \sum_{i,z} r_i G_i P_{i,z}$$

(8)

Rate of change of nutrient concentration $N$ of nutrient on patch $z$, with global turnover rate $D = 0.25$, determining the rate at which nutrients are refreshed and the nutrient supply concentration $S$.

**Generating landscapes**. We generated different fragmented landscapes, represented by random geometric graphs, by randomly drawing the locations of $Z$ patches from a uniform distribution between 0 and 1 for x- and y-coordinates, respectively.

**Dispersal**. We model dispersal between local communities as a dynamic process of emigration and immigration, assuming dispersal to occur at the same timescale as the local population dynamics. Thus, biomass flows dynamically between local populations and the dispersal dynamics directly influence local population dynamics and vice versa. We model a hostile matrix between habitat patches that does not allow for feeding interactions to occur during dispersal. The total rate of emigration of animal species $i$ from patch $z$ is

$$E_{i,z} = d_{i,z}B_{i,z}$$

(9)

with $d_{i,z}$ as the corresponding per capita dispersal rate. We model $d_{i,z}$ as

$$d_{i,z} = \frac{a}{1 + e^{-b(x_i - v_{i,z})}}$$

(10)

with $a$, the maximum dispersal rate, $b = 10$, a parameter determining the shape of the dispersal rate, $x_i$, the inflection point determined by the metabolic demands per unit biomass of species $i$, and $v_{i,z}$, the net growth rate of species $i$ on patch $z$. The net growth rate consists of the biomass gain by feeding, the biomass loss by being fed upon and the metabolic loss ($v_{i,z} = \frac{B_{i,z}\sum_j e_j F_{ij,z} - \sum_j B_{j,z}F_{ji,z} - x_i B_{i,z}}{B_{i,z}}$). We chose to model $d_{i,z}$ as a function of each species' net growth rate to account for emigration triggers, such as resource availability, predation pressure, and inter- and intra-specific competition. If for example an animal species' net growth is positive, there is no need for dispersal and emigration will be low. However, if the local environmental conditions deteriorate, the growing incentives to search for a better habitat increase the fraction of individuals emigrating.

**Immigration**. The rate of immigration of biomass density of species $i$ into patch $z$ follows

$$I_{i,z} = \sum_{n \in N_z} E_{i,n}\max(1 - \delta_{i,nz}, 0)\frac{\max(1 - \delta_{i,nz}, 0)}{\sum_{m \in N_n}\max(1 - \delta_{i,nz}, 0)}$$

(11)

where $N_z$ and $N_n$ are the sets of all patches within the dispersal range of species $i$ on patches $z$ and $n$, respectively. In this equation, $E_{i,n}$ is the emigration rate of species $i$ from patch $n$, $\max(1 - \delta_{i,nz}, 0)$ is the fraction of successfully dispersing biomass, i.e. the fraction of biomass not lost to the matrix, and $\delta_{i,nz}$ is the distance between patches $n$ and $z$ relative to species i's maximum dispersal distance $\delta_i$ (see below

paragraph Maximum dispersal distance). The term $\frac{\max(1-\delta_{i,nz},0)}{\sum_{m \in N_n} \max(1-\delta_{i,nz},0)}$ determines the fraction of biomass of species $i$ emigrating from source patch n towards target patch z. This fraction depends on the relative distance between the patches, $\delta_{i,nz}$, and the relative distances to all other potential target patches m of species $i$ on the source patch n, $\delta_{i,nm}$. Thus, the flow of biomass is greatest between patches with small distances to account for the logic that the first patch dispersing organism come across is closer. In other words, the further a destination is, the more likely it is to come across another patch before.

For numerical reasons, we did not allow for dispersal flows with $I_{i,z} < 10^{-10}$. In this case, we immediately set $I_{i,z}$ to 0. We assume that the maximum dispersal distance $\delta_i$ of animal species increases with their body mass. For animal species, the body mass $m_i$ determines how far they can travel through the matrix. Thus, animal species at high trophic positions can disperse further than smaller animals at lower trophic levels. Each animal species perceives its own dispersal network dependent on its species-specific maximum dispersal distance

$$\delta_i = \delta_0 m_i^\epsilon \qquad (12)$$

where the exponent $\epsilon = 0.05$ determines the slope of the body mass scaling of $\delta_i$ and intercept $\delta_0 = 0.1256$. This intercept had been chosen because an organism with a body mass of $10^{12}$ would have a maximum dispersal range of 0.5. We chose a positive value for $\epsilon$ to account for a higher mobility of animals with larger body masses.

**Setup**. To answer our questions, we model the following scenarios:

Nutrient enrichment (Fig. 2a): Simulations across a gradient of nutrient supply concentrations (0, 10) on one patch without emigration and therefore also no dispersal loss.

Drainage effect (Fig. 2b): Simulations across a gradient of maximal emigration rates (0, 0.15) on one eutrophic patch with a nutrient supply concentration of 10.

Hostility effect with two patches: Simulations across a gradient of dispersal losses (0, 1) on two eutrophic patches with a nutrient supply concentration of 20 on each and a maximal dispersal rate of 0.05.

Heterogeneity effect with two patches (Fig. 3, along the x-axis): Simulations across a gradient of nutrient supply concentrations (0, 20) on one of two patches with the other patch being a eutrophic patch with a nutrient supply concentration of 20, a maximal emigration rate of 0.05 and no dispersal loss.

Interaction of hostility effect and heterogeneity effect (Fig. 3): For each level of heterogeneity (difference in nutrient supply between the two patches) we simulated the whole gradient of the hostility effect (dispersal loss of the predator from 0 to 1).

Heterogeneity effect on complex food webs in complex landscapes (Fig. 4): For a complex meta-food-web, we generated five random geometric graphs consisting of 50 patches. Each patch was initialised with a complex food web consisting of 10 plant and 30 animal species. For each random geometric graph, we simulated 15 homogeneous landscapes, where all patches have the same nutrient supply concentration with simulations across a gradient of nutrient supply concentrations ranging from $10^{-0.8}$ (oligotrophic) to $10^2$ (eutrophic) in steps of 0.2 in the exponent, and 25 heterogeneous landscapes for each of the three heterogeneity scenarios (heterogeneous oligo-, meso-, and eutrophic), where the nutrient supply concentration for each patch is assigned randomly from the same gradient as in the homogeneous scenario. For the heterogeneous mesotrophic landscapes, the gradient from $10^{-0.8}$ to $10^2$ (in steps of 0.2 in the exponent) was sampled from a uniform distribution, while for the heterogenious oligotrophic and eutrophic landscapes, 42 nutrient supply values were sampled from the lower ($10^{-0.8}$ to $10^0$) or higher third ($10^{1.2}$ to $10^2$) of the gradient, repectively, 5 from the intermediate third ($10^{0.2}$ to $10^1$) and 3 of the remaining third of the gradient.

**Sensitivity analyses**. Additional results from simulations with a non-adaptive dispersal model and non-body mass-scaled dispersal ranges of organisms are presented in the Supplementary Notes. Results are, however, qualitatively the same.

**Reporting summary**. Further information on research design is available in the Nature Research Reporting Summary linked to this article.

## Data availability
All data generated in this study can be reproduced with the model code. The processed data can be found in the same github repository as the code: https://github.com/RemoRyser/Metafoodweb[41].

## Code availability
The code to reproduce the data can be found at: https://github.com/RemoRyser/Metafoodweb[41].

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

## Acknowledgements

We gratefully acknowledge the support of the German Centre for Integrative Biodiversity Research (iDiv) Halle-Jena-Leipzig funded by the German Research Foundation (FZT 118) and funding by the German Research Foundation (DFG) in the framework of the research units FOR 1748 (BR 2315/16-2) and FOR 2716 (BR 2315/21-1).

## Author contributions

R.R., U.B., and M.R.H. developed the idea, R.R. built the model with contributions by J.H., R.R. did the analyses and wrote the first draft of the manuscript with contributions by U.B., M.R.H., and D.G. M.R.H. designed Figs. 1 and 2. All authors contributed to the interpretation and the final version of this manuscript.

## Funding

## Competing interests

The authors declare no competing interests.
