## [Peer Review File · Nature Communications]

Reviewers' Comments:

Reviewer #1:

Remarks to the Author:

Remo Ryser et al, Landscape heterogeneity buffers biodiversity of meta-food-webs under global change through rescue and drainage effects.

This work looked at the interactive impact of habitat fragmentation and eutrophication on biodiversity. Since experimental studies on this topic is a challenge, they developed a meta-food-web model to simulate this question. In particular, they tried to address this question in large spatial networks with many species. I believe the way they synthesized metapopulation and food-web models as a unifying principle into a meta-food-web model is novel and exciting. They analyzed population dynamics across a gradient of complexity from simple to complex systems. They showed spatial processes in heterogenous landscapes stabilize local food-web dynamics and translate into higher diversity. Also, they found landscape heterogeneity provides a buffering capacity against increasing nutrient eutrophication.

I generally think they addressed an interesting question and has the potential to make a valuable contribution to the literature. However, I have questions on some assumptions they made in the model, and I believe much stronger links to the ecology of real populations are needed. Additionally, it is unclear about the robustness of their results given different parameter values since they did not perform sensitivity analysis.

Major comments:

I understand it is essential to make assumptions to keep the model simple, but I have some questions on assumptions they made in the model section, see comments below. I also think that the settings used in this model are simple that it remains unclear how relevant they are in real systems. In principle, it is OK to explore complex questions starting from simplified systems, but then it should be discussed more thoroughly how likely it is that the question and specific settings addressed, and the insights gained, are relevant in reality. It is important for the authors to discuss a little more why they made such assumptions, how their assumptions respect real conditions in nature, and how different assumptions will alter their results.

319 – 320. I understand one novelty of this work is to look at large spatial networks, therefore I am wondering if defining a landscape with one or two patches is enough. Or at least, they could discuss whether adding more patches will make any difference to their results.

I feel it is ok to use body masses to determine not only trophic links and interaction strengths of the food webs but also the dispersal ranges. Nonetheless, I think a stronger statement on why choosing body mass instead of other factors is needed.

I see they did not perform sensitivity analysis, so I am curious about the robustness of their results given different parameter values.

I understand testing their theory in real experiments would be difficult, but I feel they have not connected their results strong enough to real nature.

Figures are well made, very clear and easy to understand.

Minor comments

Lines 46 – 39 is not clear to me, would suggest rephrasing this sentence to make it easier to understand.

74. Consumer dispersal, is not too clear to me what it meant.

227 – 230. I am not sure if scaling up these findings to higher levels is flexible, especially since many other confounding factors were not considered in this work.

301. I am curious why using type II functional response.

316 – 317. I feel this assumption depends on the dispersal mode, so I am not sure if I agree with this assumption.

353 – 354. I don't quite understand this sentence.

Reviewer #2:

Remarks to the Author:

This paper uses a previously established size-structured meta-food web model to determine how food web stability and species persistence are affected by eutrophication (measured as rate of nutrient inputs), habitat fragmentation (measured as interpatch distances) and habitat heterogeneity (measured as variability in nutrient inputs between patches). The authors describe two mechanisms by which habitat fragmentation and heterogeneity can interact to increase biodiversity: the first is the rescue effect, where asynchronous fluctuations between disconnected patches can rescue low-density populations. The second mechanism is what the authors refer to as the drainage effect, where growth-rate dependent emigration rates reduce high-density populations, reducing large-scale enrichment-driven oscillations and preventing extinction of high-trophic level species.

I think this paper is making an important contribution to the growing meta-food-web literature. The drainage effect is an extremely interesting mechanism for stabilizing predator-prey systems, and the authors effectively demonstrate that it arises naturally from growth-rate dependent dispersal rates. As far as I know this is the first time such a stabilizing mechanism has been described for metacommunities, although Ruxton and Rohani (1999) noted a similar stabilizing effect due to fitness-dependent dispersal. I think this mechanism is a very important finding, and one that could be emphasized more strongly in the abstract.

I did have some major concerns with the model used for the paper, though. The model made several strong assumptions about species interactions that were, in my opinion, not well-justified in text or based on well-understood population dynamics. I have noted specific issues with some of the modelling choices regarding the functional responses below, but I think they are substantial enough that I am not convinced that the multispecies results in figure 4 would be robust to alternative model specifications, or that the chosen parameter values make biological sense. I think the tri-trophic results (figures 2 and 3) should be robust, but it would have helped to have the full tri-trophic food web written out as equations or to have the model code available, as it would be much easier to evaluate if the chosen parameters were reasonable. It is difficult as is to evaluate if the orders of magnitude of the different parameters in the tri-trophic model make sense as an ecological system without having the equations available in one place.

The first issue with the functional response used (line 47 in the supplement) is the weighting term (w_i), where the attack rates are weighted by $1/\text{the number of prey species}$. This is justified in text by noting that a consumer has to split its consumption between more than one resource species. However, that is already factored in to the multi-species functional response. This choice of weighting factor means that if you split a single species, R , into two equally size subgroups, $R_1 = R_2 = R/2$, the total feeding rate of the predator on the two subgroups $F_1 + F_2$ would be $(1/2bR_1 + 1/2bR_2)/(1 + 1/2bhR_1 + 1/2bhR_2) = (bR/2)/(1 + bhR/2)$. This is approximately half the total feeding rate if the two subgroups were treated as one group: $bR/(1 + bhR)$. This violates the "common sense" assumption that population dynamics shouldn't be affected by splitting a resource into ecologically equivalent sub-populations (Rossberg 2013), and effectively decreases predation rates as species diversity increases.

I also did not understand the justification for the choice of attack rate scaling with prey biomass. While it makes sense that predator attack rates should scale with predator movement speeds, attack rates should if anything decline with prey movement rates, as the prey should be more able to avoid predators the more mobile they are. The best current meta-analysis of attack rates that I'm aware of (Rall et al. 2012) showed that attack rates showed very limited scaling with resource mass. The allometric scaling coefficients for handling times in this paper were taken from Rall et al., and I did not understand why the scaling coefficients for attack rates from that paper were not used. I would recommend at least testing if removing the weighting factors and using the Rall attack rate allometric parameters makes a difference in the model results.

References:

- Rall, B.C., Brose, U., Hartvig, M., Kalinkat, G., Schwarzmüller, F., Vucic-Pestic, O., et al. (2012). Universal temperature and body-mass scaling of feeding rates. *Philos Trans R Soc Lond B Biol Sci*, 367, 2923–2934.
- Rossberg, A.G. (2013). *Food webs and biodiversity: Foundations, models, data*. John Wiley & Sons, River Street, Hoboken, NJ.
- Ruxton, G.D. & Rohani, P. (1999). Fitness-dependent dispersal in metapopulations and its consequences for persistence and synchrony. *Journal of Animal Ecology*, 68, 530–539.

Specific comments:

Line 41: I am not convinced that landscape hostility is the best term for the concept the authors are trying to convey; landscape hostility does not seem to be used as a term except in a few other papers, and it seems to be effectively synonymous with dispersal mortality. I would recommend using dispersal mortality instead, as it will be easier for researchers working on these problems to find this paper.

Line 64: “Weak biomass fluxes can cause consumer extinction due to energy limitations while strong biomass fluxes” It would be useful to clarify here whether these are weak or strong fluxes as measured in absolute terms, or as per-capita effects.

Line 72: who “They” is unclear here. I would suggest “These studies showed”.

Line 98 (Figure 1): The drainage effect is mentioned in Figure 1c), but has not yet been defined in the paper. Given its key role in the paper’s results, it should be clearly defined earlier in the manuscript.

Line 110: It may be useful to note that the assumed emigration rate is a form of fitness-dependent dispersal. In that context, Ruxton and Rohani noted an effect very similar to the indicated drainage effect, where fitness-dependent dispersal can stabilize consumer-resource cycles.

Figure 2B): Does the existence of a positive equilibrium and an extinction equilibrium at high emigration rates in the “extinction” region denote a bistable part of the parameter space? This seems like it is worth noting, as if it is bistable, it means that at high emigration rates, the system will not necessarily go deterministically extinct.

Line 138: “a gradient of emigration rates”: this should be “maximum emigration rates”, since this emigration rate will only be achieved when per-capita growth rates are strongly negative.

Line 141: The landscape heterogeneity measure here seems to be confounding the effects changing mean nutrient fluxes with altering how homogeneous the landscape is; for the heterogeneous landscape, one patch always has nutrient flux equal to 15, and the other varied in nutrient flux from zero to 15. Therefore, mean nutrient fluxes across all patches also increased with homogeneity. As such, the variation in oscillations in figure 3b may simply be a paradox of enrichment effect. I would suggest parameterizing homogeneity by setting nutrient flux rates in one patch to $7.5 + \delta$, and the other to $7.5 - \delta$, then vary δ from 0 (homogenous landscape) to 1 (totally heterogeneous).

Line 188-190: I do not understand what this means. What does sampling density mean in this context? Does this mean that multiple different homogeneous food webs were created with different nutrient supplies, but only a single heterogeneous landscape was simulated for each of the oligotrophic, mesotrophic and eutrophic cases?

Line 207: Is this alpha diversity of animals, plants, or both? It would be interesting to see if heterogeneity has any effect on plant diversity independently from animal diversity; I would

expect most of the effect to be driven by changing animal density, but indirect competitive effects may result in higher predator diversity supporting higher plant diversity.

Line 235-237: It is important to note here that this effect, of increasing dispersal mortality leading to reduced oscillations and increased persistence, is heavily dependent on the assumed dispersal model, and the assumption that fragmentation is only affecting interpatch distances (not patch sizes, or total area available to species). Connecting habitat patches via corridors not only reduces dispersal mortality, but also can increase effective patch sizes, improve gene flow, etc. It is interesting that these model results are consistent with the results in Farig (2017), showing that many species show positive relationships with increasing habitat fragmentation.

Fahrig, L. (2017). Ecological responses to habitat fragmentation per se. *Annual Review of Ecology, Evolution, and Systematics*, 48, 1–23.

Line 247-249: This is a strong statement to make based off a single model result; while this study demonstrates that drainage effects are a theoretically feasible mechanism for preventing enrichment-driven extinction, it still needs to be demonstrated that growth-rate driven dispersal is strong enough to stabilize natural populations.

Line 301: This adjustment to capture rates needs to be more clearly justified.

All further notes are on the supplemental material, referred to by line numbers S1 – S219:

Line S47: Why are these rates scaled by 1 divided by predator mass? Why not simply scale predator attack rates. Also, q is not defined here, or listed in the table of parameters. It appears to be a nonlinearity parameter to affect the shape of the functional response, but if it is not used, I would suggest removing it from the equations. In a similar vein, I would remove the $cA_{i,z}$ term here (the predator interference effect), as it is set to zero for all simulations anyway.

Line S74: “with width γ ”: given the functional response listed in equation 5, the width of the interaction kernel should scale with the inverse of γ , not proportionately to γ (i.e. a large γ value would correspond to a narrow range of interactions).

Line S115 (equation 10): There is a missing equals sign here. Also, this equation implies that dispersal rates increase with per-capita growth rates. I saw from the code with Ryser et al 2019 that the functional response actually used was $d_{iz} = a/(1 + e^{-(b(x_i - v_{iz}))})$, so this formula is missing a minus sign. However, I also cannot figure out why the term in the brackets is given as $(x_i - v_{iz})$; v_{iz} is given as the per-capita growth rate, but this is not clearly defined in the text. From the code in Ryser et al. 2019, the term v_{iz} equals the per-capita feeding rate minus the per-capita predation rate and the per-capita respiration rate (x_i). This does not explain why x_i is double-counted here when calculating the per-capita growth rate through; why not just use $-v_{iz}$ instead of $x_i - v_{iz}$?

Line S130 (equation 11): This is a somewhat odd immigration function. It implies that total dispersal success (i.e. dispersal survival) will always decline if you add an extra patch further away than the current ones. What is the justification for this? It implies that a more connected landscape is more hostile. It also has the somewhat counter-intuitive effect that if all patches are the same distance from each other and have the same number of emigrants, adding an extra patch does not increase the number of immigrants arriving in each patch, so that increasing the number of patches on the landscape does not decrease dispersal mortality.

Table TS1: What are the units for these parameters? What is the assumed time scale for the simulations?

Line S165: “Increased dispersal loss (hostility)”: it should be made clear here that dispersal loss is varied by altering inter-patch distances. This also means that hostility will vary with predator body mass, as larger predators are assumed to have higher dispersal distances, and thus less dispersal

mortality at any given distance.

Reviewer #3

The manuscript studies a two-patch food-chain model and a multi-patch food-web model to explore how the metacommunity structure affects the destabilizing roles of eutrophication in food-web dynamics. The authors identify the drainage effect, which means the diversion of top predator biomass from enriched local food webs and acts to stabilize food-web dynamics by weakening top-down suppression. The diverted biomass is lost in the uninhabitable landscape matrix or moves to oligotrophic patches.

I think the setup of the manuscript is timely in the sense that it aims at clarifying the consequences and management of multiple environmental stressors in ecological systems. However, its conceptual contributions to ecology are not clearly presented or, in my opinion, weak. Moreover, explanations on model details are insufficient in many places, which prevents precise reading. Finally, some comparisons between different treatments in simulation experiments seem inadequate. Below I summarize my comments. I have also added notes directly on the manuscript and supplement files.

- (1) The manuscript does not fully explain the underlying behavioral mechanisms for between-patch movements, although these mechanisms are the main drivers of how biomass diversion occurs and the drainage effect emerges. If I understand correctly, organisms are more likely to leave the current patches when the rate of net biomass change due to trophic interactions (increase due to prey consumption and decrease due to being predated) is low. For top predators, they become more likely to leave their current patches when their prey is scarce. Prey becomes scarce, when eutrophication increases predator biomass and increased predators in turn reduce prey biomass. Predator emigration from such occasions will save prey from predator's runaway consumption, and thus stabilize the local dynamics. In my opinion, this effect is not entirely new and may be related to dispersal-induced indirect negative density dependence. There is rich literature on the effects of spatial processes on the stability of predator-prey dynamics. For example, Gounand et al. (2014, *Am Nat*, cited in the manuscript) provides a concise review in their introduction section. I recommend that the authors relate the mechanisms driving the drainage effect in an adequate broader conceptual framework, and explain what parts of their findings are essentially novel.
- (2) In the current model framework, the probability that organisms leave the current patch depends on the conditions, such as their prey and predator biomasses, determining their local population growth rates. Although this may be a reasonable assumption, other factors can also affect the probability of immigration. For example, organisms may take into account the cost of immigration (possible mortality during dispersal in the landscape matrix) to decide whether to disperse. This can weaken the drainage effect because predators in eutrophic patches may become less likely to leave the patches and die in the landscape matrix. Moreover, passive

dispersal may be another possible assumption. I suggest that the authors test and compare different rules of dispersal decision and how they affect the stabilizing drainage effect. This is important for evaluating the general importance of the drainage effect.

- (3) Although the authors focus on the stabilizing drainage effect resulting from the diversion of predator biomass, I wonder how movements of prey biomass can have interacting roles in the stabilizing drainage effects. The rate of prey immigration depends on the per capita biomass growth determined by resource acquisition and loss by predation. This means that prey is more likely to leave patches when they suffer from high predation. This might help sustain prey population and be stabilizing, decrease their predator's resource and be destabilizing, or impose further dispersal mortality on prey and be destabilizing. Are there any roles of prey movements in dynamical stability, and how does the difference in dispersal spatial scales between prey and predators affect these roles?
- (4) The comparison between different landscape scenarios in figure 4 is inadequate, because total nutrient supply rates (summed across the whole landscape) is different between landscapes of different heterogeneity. The rescue effect should be evaluated by comparing an oligotrophic patch between homogeneous and heterogeneous landscapes of the same total (i.e., average) nutrient supply rate (the green double arrow in the left panel of the figure below). Similarly, the drainage effect should be evaluated by comparing a eutrophic patch between homogeneous and heterogeneous landscapes of the same total (i.e., average) nutrient supply rate (the blue double arrow in the middle panel). For the oligotrophic heterogeneous landscape scenario (the orange shadow in the right panel), alpha-diversity of any patches, irrespective of their local nutrient supply rates, is greater than that of a patch in the homogeneous landscape with the same average nutrient supply rate. This might be because both rescue and drainage effects are effective and strengthen each other.

I assume that middle bars in the violin plots are the averages.

REVIEWER COMMENTS

Reviewer #1 (Remarks to the Author):

Remo Ryser et al, Landscape heterogeneity buffers biodiversity of meta-food-webs under global change through rescue and drainage effects.

This work looked at the interactive impact of habitat fragmentation and eutrophication on biodiversity. Since experimental studies on this topic is a challenge, they developed a meta-food-web model to simulate this question. In particular, they tried to address this question in large spatial networks with many species. I believe the way they synthesized metapopulation and food-web models as a unifying principle into a meta-food-web model is novel and exciting. They analyzed population dynamics across a gradient of complexity from simple to complex systems. They showed spatial processes in heterogeneous landscapes stabilize local food-web dynamics and translate into higher diversity. Also, they found landscape heterogeneity provides a buffering capacity against increasing nutrient eutrophication.

I generally think they addressed an interesting question and has the potential to make a valuable contribution to the literature. However, I have questions on some assumptions they made in the model, and I believe much stronger links to the ecology of real populations are needed. Additionally, it is unclear about the robustness of their results given different parameter values since they did not perform sensitivity analysis.

Major comments:

I understand it is essential to make assumptions to keep the model simple, but I have some questions on assumptions they made in the model section, see comments below. I also think that the settings used in this model are simple that it remains unclear how relevant they are in real systems. In principle, it is OK to explore complex questions starting from simplified systems, but then it should be discussed more thoroughly how likely it is that the question and specific settings addressed, and the insights gained, are relevant in reality. It is important for the authors to discuss a little more why they made such assumptions, how their assumptions respect real conditions in nature, and how different assumptions will alter their results.

We appreciate this comment, and we agree that modeling results generally need to be validated empirically. Simultaneously, empirical experiments on such a complexity scale are very challenging and may only be feasible with, for example, microbial systems. In general, our model's terms and parameters are chosen that they are consistent with prior work (e.g., Schneider et al. 2016, Ryser et al. 2019). This consistency facilitates comparisons across studies. We also performed additional simulations as outlined in the general response to generalize our results regarding our assumptions. We also added a few sentences in the discussion addressing this (new line: 282).

319 – 320. I understand one novelty of this work is to look at large spatial networks, therefore I am wondering if defining a landscape with one or two patches is enough. Or at least, they could discuss whether adding more patches will make any difference to their results.

Indeed, we did not explicitly mention our simulations of complex landscapes comprising 50 patches in this sentence. We appreciated this clarification of our approach (new lines:330 & 369).

I feel it is ok to use body masses to determine not only trophic links and interaction strengths of the food webs but also the dispersal ranges. Nonetheless, I think a stronger statement on why choosing body mass instead of other factors is needed.

We agree with this point and further emphasized the importance of body mass for dispersal ranges (line 91). Additionally, we performed new simulations with equal dispersal ranges of all species that show our results' robustness (in the Supplement Fig.S5). Finally, we discussed the implications of varying dispersal modes (e.g. flying) (new line: 280).

I see they did not perform sensitivity analysis, so I am curious about the robustness of their results given different parameter values.

We appreciate and understand the interest in sensitivity analyses. We did additional simulations with different dispersal models (Supplement Figures S3-S5), differently parameterized functional responses (Figure below in the response), and a landscape heterogeneity manipulation independent of the landscape level nutrient supply (Supplement Fig.S6). As outlined in our general response regarding sensitivity analyses above, the main manuscript results are robust against these variations.

I understand testing their theory in real experiments would be difficult, but I feel they have not connected their results strong enough to real nature.

We followed this suggestion and elaborated on this further in the discussion (lines 291 and following). We discuss in what framework our modeling approach could be tested empirically and where the modeling approach may remain the only option.

Figures are well made, very clear and easy to understand.

We appreciate this positive feedback as we have been putting a lot of time and effort into the figures.

Minor comments

Lines 46 – 39 is not clear to me, would suggest rephrasing this sentence to make it easier to understand.

Rephrased

74. Consumer dispersal, is not too clear to me what it meant.

Changed to dispersal of consumers

227 – 230. I am not sure if scaling up these findings to higher levels is flexible, especially since many other confounding factors were not considered in this work.

Replaced "findings" with "processes" – it was meant to be an outlook that highlights that these two approaches could be combined

301. I am curious why using type II functional response.

Following prior work (Schneider et al. 2016, Ryser et al. 2019, McCann et al. 2005, Brose et al. 2006), we have based our models on type-II functional responses as they are

conceptually most simple and only assume handling time and attack rates. The consistency in the model terms facilitates the comparison across studies.

316 – 317. I feel this assumption depends on the dispersal mode, so I am not sure if I agree with this assumption.

While we do not model different dispersal modes of organisms, we agree that this is interesting and may have implications for how much drainage a given organism would experience. We acknowledge this now in the discussion (line 279).

353 – 354. I don't quite understand this sentence.

This sentence gives the description of simulations along the x-axis of Fig. 3. We specified this.

Reviewer #2 (Remarks to the Author):

This paper uses a previously established size-structured meta-food web model to determine how food web stability and species persistence are affected by eutrophication (measured as rate of nutrient inputs), habitat fragmentation (measured as interpatch distances) and habitat heterogeneity (measured as variability in nutrient inputs between patches). The authors describe two mechanisms by which habitat fragmentation and heterogeneity can interact to increase biodiversity: the first is the rescue effect, where asynchronous fluctuations between disconnected patches can rescue low-density populations. The second mechanism is what the authors refer to as the drainage effect, where growth-rate dependent emigration rates reduce high-density populations, reducing large-scale enrichment-driven oscillations and preventing extinction of high-trophic level species.

I think this paper is making an important contribution to the growing meta-food-web literature. The drainage effect is an extremely interesting mechanism for stabilizing predator-prey systems, and the authors effectively demonstrate that it arises naturally from growth-rate dependent dispersal rates. As far as I know this is the first time such a stabilizing mechanism has been described for metacommunities, although Ruxton and Rohani (1999) noted a similar stabilizing effect due to fitness-dependent dispersal. I think this mechanism is a very important finding, and one that could be emphasized more strongly in the abstract.

We thank the reviewer for this positive comment and highlight the drainage effect more strongly in the abstract and the introduction.

I did have some major concerns with the model used for the paper, though. The model made several strong assumptions about species interactions that were, in my opinion, not well-justified in text or based on well-understood population dynamics. I have noted specific issues with some of the modelling choices regarding the functional responses below, but I think they are substantial enough that I am not convinced that the multi-species results in figure 4 would be robust to alternative model specifications, or that the chosen parameter values make biological sense. I think the tri-trophic results (figures 2 and 3) should be robust, but it would have helped to have the full tri-trophic food web written out as equations or to have the model code available, as it would be much easier to evaluate if the chosen parameters were reasonable. It is difficult as is to evaluate if the orders of magnitude of the different parameters in the tri-trophic model make sense as an ecological system without having the equations available in one place.

We followed this suggestion by (1) including more justifications for parameter choices and (2) clarifying their assumptions in the methods section and the Supplement. Also, we added the general equations with units to the methods. For our response concerning the multi-species functional responses, we refer to the subsequent responses regarding "omega" and the encounter rates.

The first issue with the functional response used (line 47 in the Supplement) is the weighting term (w_i), where the attack rates are weighted by 1/the number of prey species. This is justified in text by

noting that a consumer has to split its consumption between more than one resource species. However, that is already factored in to the multi-species functional response. This choice of weighting factor means that if you split a single species, R , into two equally size subgroups, $R_1 = R_2 = R/2$, the total feeding rate of the predator on the two subgroups $F_1 + F_2$ would be $(1/2bR_1 + 1/2bR_2)/(1 + 1/2bhR_1 + 1/2bhR_2) = (bR/2)/(1 + bhR/2)$. This is approximately half the total feeding rate if the two subgroups were treated as one group: $bR/(1 + bhR)$. This violates the "common sense" assumption that population dynamics shouldn't be affected by splitting a resource into ecologically equivalent sub-populations (Rossberg 2013), and effectively decreases predation rates as species diversity increases.

We appreciate this comment that has stimulated multiple changes in our manuscript. Firstly, the description of the weighting term w_i in our manuscript's original version was indeed inaccurate. This term w_i can be understood as an inefficiency term for generalists (Supplement, line 50). While this is a straightforward relationship, it arises from the assumption that a generalist is less adapted to hunting a specific prey species. Instead, it is more adapted to hunting multiple prey species, rendering its attack less effective on a particular prey species. However, we agree that this formulation of generalists' inefficiency could be improved in the future. Secondly, to our knowledge, we used a "state of the art" multi-species functional response that in itself could also be improved in the future. For instance, the allometric scaling of attack rates and handling types implies that prey species pools do not have a joint density. This is also especially true for nonlinearities in density dependence. Biologically speaking, this may be true for predators that have a prey-specific search pattern or hunting strategy that would imply it cannot search for prey of different species simultaneously. The given example of dividing a population into two spatiotemporally co-occurring subpopulations would, in this case, also not result in identical consumption rates. Thus, a necessary improvement of functional responses relies on clear species definitions that are assumed to be functionally different. Generalizing the multi-species functional response to relax some of these assumptions would be an exciting research topic but beyond this work's scope. However, we carried out some additional simulations of the complex food web in heterogeneous mesotrophic landscapes (see figure below), where we removed the w_i as suggested in the comment above. This change resulted in qualitatively similar results, but it increased the persistence of the food web slightly. This finding may indeed point towards the conclusion that generalists are strongly punished in our model. As these additional simulations also show that the results shown in our manuscript are qualitatively robust against these changes in the functional response, we have been keeping our standard version of the multi-species functional response (as it is consistent with prior publications). We highly appreciate this discussion and will work on an improved functional response model in future projects.

Fig: New simulations for heterogeneous mesotrophic landscapes without w_i (dashed line) and with w_i (solid line; as in the manuscript).

I also did not understand the justification for the choice of attack rate scaling with prey biomass. While it makes sense that predator attack rates should scale with predator movement speeds, attack rates should if anything decline with prey movement rates, as the prey should be more able to avoid predators the more mobile they are. The best current meta-analysis of attack rates that I'm aware of (Rall et al. 2012) showed that attack rates showed very limited scaling with resource mass. The allometric scaling coefficients for handling times in this paper were taken from Rall et al., and I did not understand why the scaling coefficients for attack rates from that paper were not used. I would recommend at least testing if removing the weighting factors and using the Rall attack rate allometric parameters makes a difference in the model results.

We thank the reviewer for this comment, and we specified our assumption more clearly (Supplement, line 62). In our model, we have disentangled the processes of predator-prey encounter and attack. The allometric scaling of predator and prey speed referred to above influences their encounter rates. This choice is based on the simple assumption that increased predator and prey movement should result in higher encounter probabilities. After an encounter, we model the capture success, formulated in L_{ij} , as a unimodal model across the body-mass axis. This unimodal model includes the negative effect of increasing prey escape speed (mentioned in the comment) along the body mass axis on the attack rates. In contrast to the macroecological scaling relationships across predator species by Rall et al. (2012), this separation of encounter and attack has two advantages. First, it provides mechanistic models of attack rates empirically based on the allometric scaling of speed. Second, the unimodal models of attack success allow consistent modeling of food-web structure (who eats whom) and food-web dynamics (attack rates).

References:

Rall, B.C., Brose, U., Hartvig, M., Kalinkat, G., Schwarzmüller, F., Vucic-Pestic, O., et al. (2012). Universal temperature and body-mass scaling of feeding rates. *Philos Trans R Soc Lond B Biol Sci*, 367, 2923–2934.

Rossberg, A.G. (2013). *Food webs and biodiversity: Foundations, models, data*. John Wiley & Sons, River Street, Hoboken, NJ.

Ruxton, G.D. & Rohani, P. (1999). Fitness-dependent dispersal in metapopulations and its consequences for persistence and synchrony. *Journal of Animal Ecology*, 68, 530–539.

Specific comments:

Line 41: I am not convinced that landscape hostility is the best term for the concept the authors are trying to convey; landscape hostility does not seem to be used as a term except in a few other papers, and it seems to be effectively synonymous with dispersal mortality. I would recommend using dispersal mortality instead, as it will be easier for researchers working on these problems to find this paper.

We agree that dispersal mortality is easier to understand – we thus added clarifications in the introduction and the results. However, we chose the term dispersal loss as it also includes the association of metabolic demands during dispersal. Our model works with biomass densities, and we think that the term dispersal mortality is too strongly associated with individuals.

Line 64: "Weak biomass fluxes can cause consumer extinction due to energy limitations while strong biomass fluxes" It would be useful to clarify here whether these are weak or strong fluxes as measured in absolute terms, or as per-capita effects.

We clarified the sentence (line 60). It is weak and strong fluxes relative to losses.

Line 72: who "They" is unclear here. I would suggest "These studies showed".

Done

Line 98 (Figure 1): The drainage effect is mentioned in Figure 1c), but has not yet been defined in the paper. Given its key role in the paper's results, it should be clearly defined earlier in the manuscript.

We emphasized the drainage effect more in the introduction and the abstract.

Line 110: It may be useful to note that the assumed emigration rate is a form of fitness-dependent dispersal. In that context, Ruxton and Rohani noted an effect very similar to the indicated drainage effect, where fitness-dependent dispersal can stabilize consumer-resource cycles.

We appreciate this input but refrained from focusing on the adaptive dispersal as the additional simulations with the non-adaptive dispersal model produce the same effect.

Figure 2B): Does the existence of a positive equilibrium and an extinction equilibrium at high emigration rates in the "extinction" region denote a bistable part of the parameter space? This seems like it is worth noting, as if it is bistable, it means that at high emigration rates, the system will not necessarily go deterministically extinct.

This is indeed the case. We modified the figure to show this.

Line 138: "a gradient of emigration rates": this should be "maximum emigration rates", since this emigration rate will only be achieved when per-capita growth rates are strongly negative.

True – adjust figure legend

Line 141: The landscape heterogeneity measure here seems to be confounding the effects changing mean nutrient fluxes with altering how homogeneous the landscape is; for the heterogeneous landscape, one patch always has nutrient flux equal to 15, and the other varied in nutrient flux from zero to 15. Therefore, mean nutrient fluxes across all patches also increased with homogeneity. As

such, the variation in oscillations in figure 3b may simply be a paradox of enrichment effect. I would suggest parameterizing homogeneity by setting nutrient flux rates in one patch to $7.5 + \delta$, and the other to $7.5 - \delta$, then vary δ from 0 (homogenous landscape) to 1 (totally heterogeneous).

Indeed, we present the drainage effect from a patch's perspective and not on a landscape level. Scaling this up to a landscape level would require knowledge of interaction effects of the rescue and the drainage effect, which seems to be quite tricky in complex systems with several patches. Nonetheless, we performed additional simulations, where we kept the total nutrient availability in the whole landscape constant (Fig. S6 in the Supplement). This figure shows that the effect of heterogeneity in reducing oscillation amplitudes remains, although somewhat weaker, when heterogeneity is modified independently from the total nutrient availability. The reason the effect remains is most likely because biomass densities do not increase linearly with nutrient supply. Throughout the revised manuscript, we highlighted that we demonstrate the drainage effect from a patch perspective (for instance, line 161). Please also see the response below (reviewer 3, point 4).

Line 188-190: I do not understand what this means. What does sampling density mean in this context? Does this mean that multiple different homogeneous food webs were created with different nutrient supplies, but only a single heterogeneous landscape was simulated for each of the oligotrophic, mesotrophic and eutrophic cases?

We appreciate this comment and clarified our approach (lines 189 & 432). Five different landscape structures (x/y-coordinates for all patches and resulting distances) were created. In the homogeneous scenario, each of the five landscape structures was simulated at each nutrient level, resulting in 80 simulations. Then, for each of the 3 heterogeneous scenarios (heterogeneous oligo-, meso- and eutrophic) 5 sets of nutrient supply for each patch was created. Each of the 5 landscapes was simulated with each of the 5 sets of nutrient supplies in each scenario, resulting in $5 \times 5 \times 3 = 75$ simulations.

Line 207: Is this alpha diversity of animals, plants, or both? It would be interesting to see if heterogeneity has any effect on plant diversity independently from animal diversity; I would expect most of the effect to be driven by changing animal density, but indirect competitive effects may result in higher predator diversity supporting higher plant diversity.

We appreciate this suggestion, and we added figures S7-S8 in the Supplement separating the response of plants and animals. Interestingly, the direct effect of the rescue effect on animals seems to indirectly support plant diversity on oligotrophic patches. Still, the direct impact of the drainage effect on animals does not affect plant diversity a lot on eutrophic patches. Higher trophic diversity may result in better and more even control of herbivores releasing the plants from herbivory rendering them better at coping with low nutrient availability. This process may even be contra-productive on eutrophic patches, but effects on the complex dynamics seem untraceable by logic alone.

Line 235-237: It is important to note here that this effect, of increasing dispersal mortality leading to reduced oscillations and increased persistence, is heavily dependent on the assumed dispersal model, and the assumption that fragmentation is only affecting interpatch distances (not patch sizes, or total area available to species). Connecting habitat patches via corridors not only reduces dispersal mortality, but also can increase effective patch sizes, improve gene flow, etc. It is interesting that these model results are consistent with the results in Farig (2017), showing that many species show positive relationships with increasing habitat fragmentation.

In our case, we show with the additional simulations with different dispersal models that the drainage effect remains robust. We do not model patch sizes explicitly (the model uses species' densities as the state variable). Therefore, we cannot address patch area effects in the present manuscript. However, we agree that there are more facets to connectivity than just energy and biomass fluxes. Hence, we addressed differences in gene flow in the revised discussion (new line: 289).

Fahrig, L. (2017). Ecological responses to habitat fragmentation per se. *Annual Review of Ecology, Evolution, and Systematics*, 48, 1–23.

Line 247-249: This is a strong statement to make based off a single model result; while this study demonstrates that drainage effects are a theoretically feasible mechanism for preventing enrichment-driven extinction, it still needs to be demonstrated that growth-rate driven dispersal is strong enough to stabilize natural populations.

We agree and emphasized the need for empirical validation in the discussion.

Line 301: This adjustment to capture rates needs to be more clearly justified.

We added clarifications for this.

All further notes are on the supplemental material, referred to by line numbers S1 – S219:

Line S47: Why are these rates scaled by 1 divided by predator mass? Why not simply scale predator attack rates. Also, q is not defined here, or listed in the table of parameters. It appears to be a nonlinearity parameter to affect the shape of the functional response, but if it is not used, I would suggest removing it from the equations. In a similar vein, I would remove the $cA_{i,z}$ term here (the predator interference effect), as it is set to zero for all simulations anyway.

These rates are scaled by 1 divided by predator mass to transform consumption rates to consumption rates per unit biomass, which is then multiplied by the biomass density to get the rate of change in biomass density. The parameter q is listed in the table, and we further clarified the use of it. The parameter $cA_{i,z}$ is indeed not used, and we removed it.

Line S74: "with width γ ": given the functional response listed in equation 5, the width of the interaction kernel should scale with the inverse of γ , not proportionately to γ (i.e. a large γ value would correspond to a narrow range of interactions).

True, adjusted the description.

Line S115 (equation 10): There is a missing equals sign here. Also, this equation implies that dispersal rates increase with per-capita growth rates. I saw from the code with Ryser et al 2019 that the functional response actually used was $d_{iz} = a/(1 + e^{-(x_i - v_{iz})})$, so this formula is missing a minus sign. However, I also cannot figure out why the term in the brackets is given as $(x_i - v_{iz})$; v_{iz} is given as the per-capita growth rate, but this is not clearly defined in the text. From the code in Ryser et al. 2019, the term v_{iz} equals the per-capita feeding rate minus the per-capita predation rate and the per-capita respiration rate (x_i). This does not explain why x_i is double-counted here when calculating the per-capita growth rate through; why not just use $-v_{iz}$ instead of $x_i - v_{iz}$?

We appreciate this comment and specified v_{iz} . Indeed the minus sign was missing, and we corrected this. X_i is double-counted to shift the inflection point (which is at $d_{iz}=a/2$) of the function to x_i . This was chosen to have more traceable dispersal rates on patches that do not contain resources or prey, i.e. when species only experience metabolic losses. This is not rooted in a biological understanding but in foresight to potentially add empty/non-habitat patches in the model. We also added the "equals" sign.

Line S130 (equation 11): This is a somewhat odd immigration function. It implies that total dispersal success (i.e. dispersal survival) will always decline if you add an extra patch further away than the current ones. What is the justification for this? It implies that a more connected landscape is more hostile. It also has the somewhat counter-intuitive effect that if all patches are the same distance from each other and have the same number of emigrants, adding an extra patch does not increase the number of immigrants arriving in each patch, so that increasing the number of patches on the landscape does not decrease dispersal mortality.

This is an interesting discussion point as some of the properties mentioned above may appear counter-intuitive at first glance. We have phrased the model that it keeps the mass-balance across distances and the number of patches. This creates the patterns mentioned above, which we explain in the following. Every additional patch within the dispersal range of a

species provides another possible destination, thus essentially reducing the dispersing biomass from a focal patch to all others. But at the same time, the additional patch provides dispersing biomass to all others. Hence, the total biomass lost increases with every additional patch, but so does the total biomass in the metacommunity. Also, the further away an additional patch is, the less it matters, as the dispersing biomass is distributed with a weight towards the closest patches first, and then losses are calculated. Therefore, an additional patch B that is just within the dispersal range of a species on patch A results in almost all biomass dispersing to B is lost. Still, this dispersing biomass is also a tiny fraction of the total dispersing biomass from A. Last, having patches of the same distance and adding an additional patch with the same distance is only possible when increasing from 2 to 3 patches. Overall, we thus argue that our formulation of the model is reasonable and avoids violations of mass-balance.

Table TS1: What are the units for these parameters? What is the assumed time scale for the simulations?

We added the general equations with units of the general terms to the method section. The time scale is set by the metabolic rate and its empirical origin, i.e. one time step corresponds to an hour. Food chain simulations were run for 20'000 time steps and food web simulations for 100'000 time steps, corresponding to 2.28 years and 11.42 years, respectively. Scaling exponents are unitless, and intercepts have units depending on the scaling relationship but combined, resulting in meaningful units for the variables on the equation's left-hand side.

Line S165: "Increased dispersal loss (hostility)": it should be made clear here that dispersal loss is varied by altering inter-patch distances. This also means that hostility will vary with predator body mass, as larger predators are assumed to have higher dispersal distances, and thus less dispersal mortality at any given distance.

We appreciate this input. Indeed, the dispersal loss and the experienced matrix hostility depend on an animal's body mass for a given interpatch distance. We performed additional simulations with non-body-mass-dependent dispersal range and present these now in Supplement Fig. S6.

Reviewer #3 (Remarks to the Author):

The manuscript studies a two-patch food-chain model and a multi-patch food-web model to explore how the metacommunity structure affects the destabilizing roles of eutrophication in food-web dynamics. The authors identify the drainage effect, which means the diversion of top predator biomass from enriched local food webs and acts to stabilize food-web dynamics by weakening top-down suppression. The diverted biomass is lost in the uninhabitable landscape matrix or moves to oligotrophic patches.

I think the setup of the manuscript is timely in the sense that it aims at clarifying the consequences and management of multiple environmental stressors in ecological systems. However, its conceptual contributions to ecology are not clearly presented or, in my opinion, weak. Moreover, explanations on model details are insufficient in many places, which prevents precise reading. Finally, some comparisons between different treatments in simulation experiments seem inadequate. Below I summarize my comments. I have also added notes directly on the manuscript and Supplement files.

(1) The manuscript does not fully explain the underlying behavioral mechanisms for between-patch movements, although these mechanisms are the main drivers of how biomass diversion occurs and the drainage effect emerges. If I understand correctly, organisms are more likely to leave the current patches when the rate of net biomass change due to trophic interactions (increase due to prey consumption and decrease due to being predated) is low. For top predators, they become more likely to leave their current patches when their prey is scarce. Prey becomes scarce, when eutrophication increases predator biomass and increased predators in turn reduce prey

biomass. Predator emigration from such occasions will save prey from predator's runaway consumption, and thus stabilize the local dynamics. In my opinion, this effect is not entirely new and may be related to dispersal-induced indirect negative density dependence. There is rich literature on the effects of spatial processes on the stability of predator-prey dynamics. For example, Gounand et al. (2014, Am Nat, cited in the manuscript) provides a concise review in their introduction section. I recommend that the authors relate the mechanisms driving the drainage effect in an adequate broader conceptual framework, and explain what parts of their findings are essentially novel.

We performed additional simulations with different dispersal models. (1) A non-adaptive dispersal model where emigration rates do not depend on local conditions, and (2) a dispersal model with no body-mass scaling of the organisms' dispersal ranges, i.e., all organisms have the same dispersal range. We outline these in more detail in the general response and present the results in the Supplement. They show that the general patterns and the drainage effect remain the same.

It is true that the drainage effect is related to indirect negative density dependence. We included further citations and more explicitly referred to this in the introduction and the discussion. However, we think that we conceptualize and show the effects of ecologically realistic drivers on the magnitude of dispersal fluxes.

The second part of this comment raises concerns about whether the results presented in our manuscript are sufficiently novel. We feel that this is always a difficult discussion as this evaluation is quite subjective. While our approach has profited from the results published by Gounand et al. (2014), we see substantial novelty in our findings, including (1) increasing the complexity of the community from food chains to food webs, (2) addressing the consequences of heterogeneity in large spatial networks, (3) including several more realistic dispersal models such as the species-specific dispersal networks used in the main part of our manuscript, and (4) the development of an explicit concept of the drainage effect. While we appreciate this clarification, we think that our study goes beyond simple negative density dependence with these points.

(2) In the current model framework, the probability that organisms leave the current patch depends on the conditions, such as their prey and predator biomasses, determining their local population growth rates. Although this may be a reasonable assumption, other factors can also affect the probability of immigration. For example, organisms may take into account the cost of immigration (possible mortality during dispersal in the landscape matrix) to decide whether to disperse. This can weaken the drainage effect because predators in eutrophic patches may become less likely to leave the patches and die in the landscape matrix. Moreover, passive dispersal may be another possible assumption. I suggest that the authors test and compare different rules of dispersal decision and how they affect the stabilizing drainage effect. This is important for evaluating the general importance of the drainage effect.

We appreciate this input and agree that there are many options to model the dispersal. As outlined in the general response, we performed additional simulations with two different dispersal functions: one with non-adaptive dispersal and one without body-mass scaled dispersal ranges. Our results remained robust across all three dispersal models (the original and the two additional models). However, all of the used dispersal models are non-informed, i.e. dispersing organisms have no information on their dispersal destiny conditions. While we agree that this could be very interesting, we also see that including the use of information into dispersal models and addressing this issue is beyond this manuscript's scope.

(3) Although the authors focus on the stabilizing drainage effect resulting from the diversion of predator biomass, I wonder how movements of prey biomass can have interacting roles in the stabilizing drainage effects. The rate of prey immigration depends on the per capita biomass growth determined by resource acquisition and loss by predation. This means that prey is more likely to leave patches when they suffer from high predation. This might help sustain prey population and be stabilizing, decrease their predator's resource and be destabilizing, or impose further dispersal mortality on prey and be destabilizing. Are there any roles of prey movements in dynamical stability, and how does the difference in dispersal spatial scales between prey and predators affect these roles?

As outlined in the general response, the additional sensitivity analyses with different dispersal models did not qualitatively change the results. However, removing the body-mass scaled dispersal range, i.e. no difference in dispersal distances between prey and predators, did shift the oscillation pattern compared to Fig.3 up a bit along the y-axis. This shift can be explained by the fact that the prey's dispersal loss is slightly lower, and the overall drainage effect is weakened. This suggests that the drainage effect is mainly driven by flows in predator biomass and to a lesser degree by flows in prey biomass. Nevertheless, we agree that disentangling the direct effects of dispersal flows from indirect effects via induced changes in the interacting species' density would be interesting. Still, a more detailed analysis of such a connection of direct dispersal effects with potential trophic cascades goes beyond this manuscript's scope.

(4) The comparison between different landscape scenarios in figure 4 is inadequate, because total nutrient supply rates (summed across the whole landscape) is different between landscapes of different heterogeneity. The rescue effect should be evaluated by comparing an oligotrophic patch between homogeneous and heterogeneous landscapes of the same total (i.e., average) nutrient supply rate (the green double arrow in the left panel of the figure below). Similarly, the drainage effect should be evaluated by comparing a eutrophic patch between homogeneous and heterogeneous landscapes of the same total (i.e., average) nutrient supply rate (the blue double arrow in the middle panel). For the oligotrophic heterogeneous landscape scenario (the orange shadow in the right panel), alpha-diversity of any patches, irrespective of their local nutrient supply rates, is greater than that of a patch in the homogeneous landscape with the same average nutrient supply rate. This might be because both rescue and drainage effects are effective and strengthen each other.

We highly appreciate this input and find it very interesting to think of how rescue and drainage effects indirectly cascade through the food webs and the spatial networks. In this manuscript, however, we focus on rescue and drainage effects at the spatial scale of patches. Thus, we argue that the comparison of single patches embedded in landscapes is adequate in this context as it is our study's explicit aim. However, we agree that our focus on the term "heterogeneity" might have been oversimplifying the discussion as it is indeed in many landscape configurations correlated with changes in landscape eutrophication. In the revised version of our manuscript, we have mentioned this correlation and stated that "Unbiased comparisons across landscapes should thus be restricted to the same landscape average of nutrient supply (diamonds in the violin plots relative to purple line of Fig. 4)." (lines 244-247).

The thoughtful additions by Reviewer 3 to our graph are exciting and are fuel for future research. In Fig. 4 of our manuscript, we present results at the spatial scale of patches to show that an oligotrophic patch can benefit a eutrophic patch (drainage effect) and vice versa (rescue effect). The green and blue arrows and the orange area added to the figure below by Reviewer 3 integrate our patch-level concept with landscape-level patterns. Despite our interest in this topic, we caution that such integration across spatial scales would also demand incorporating indirect effects cascading across the spatial networks of patches that also depend on patch locations and distances (lines 248-250). A systematic exploration of such effects across spatial scales exceeds the computational capacity of the current project. As the comparisons between the landscape scenarios at the same level of total nutrient supply (positions on the x-axis indicated by the diamonds in the violin plot of Fig. 4, and arrows added by Reviewer 3 to the figure below) suggest that the rescue and drainage effect hold irrespective of landscape nutrient supply, we have kept our original results and added this point to the discussion (lines 244-250). Nevertheless, we performed additional simulations of food chains on two patches to test for independent effects of heterogeneity and landscape nutrient supply. These simulations demonstrate an effect of heterogeneity that is independent of landscape nutrient supply (see Fig. S6 in the Supplement).

I assume that middle bars in the violin plots are the averages.

Minor comments in the manuscript:

Minor comments provided by reviewer 3 have been incorporated in the manuscript and responses are provided whenever necessary.

L121: Explanation is insufficient. Explain a concrete logic conveyed by the "turning around ..." argument. Moreover, why is this drainage effect "additional?"

Additional, in the sense of adding spatial processes to trophic processes. Rephrased to "spatial energy transfer"

L152: The drainage effect here is not clear. Does it mean the reduction of predator biomass from eutrophic patches (the effect is on the local patch scale), or the reduction of total predator biomass across the metacommunity (the effect on the regional scale).

We added a definition of the drainage effect. We look at it (and define it) from the perspective of a single patch. See general response also.

Reviewers' Comments:

Reviewer #1:

Remarks to the Author:

I appreciate all the efforts that the authors have made to improve their work, the manuscript looks much better and stronger now.

Reviewer #2:

Remarks to the Author:

This is my second time reviewing this paper. The authors have thoughtfully responded to most of my and the other reviewer's comments on the paper. I do still have some issues with how nutrient heterogeneity is being included in the model, as well as with how well the drainage effect has been connected to prior research on stabilizing mechanisms in food webs.

My primary concern is still regarding the basis for how heterogeneous nutrient supply rates were included in the modelling framework. In the author's response to my initial comments, they noted that they focused on the drainage effect from a patch-perspective, which I take to mean that they focused on the effect of the drainage effect on the eutrophic patch. I agree that this is a reasonable framing for figure 3b. However, the focus of figure 3a) and possibly figure 4) are on the coexistence of multiple species across a landscape, which is very much not a patch-perspective, and will be heavily affected by average nutrient supplies to the entire landscape. Figure S6 seems to imply that fluctuation magnitudes are not that sensitive to heterogeneity when total input rates are kept constant. This should be something noted in the main text, as the stabilizing effects of reducing total energy inputs into a system are a well-known phenomenon.

Specific comments:

Line 24: "drainage effect' stabilizes biodiversity by more uniform distributions": this phrasing is unclear. I would rephrase this to clarify what the drainage effect is. Something along the lines of "drainage effects' due to high dispersal mortality from abundant patches stabilize biodiversity dynamics of eutrophic patches. This effect is magnified in heterogeneous landscapes."

Line 129: It would be useful to make the direct connection here that emigration from these patches in the single-patch model is effectively acting as an additional source of mortality; since the emigrants are being lost to the system, emigration is indistinguishable from mortality. Since this will reduce the average growth rate of all species, it should naturally act to stabilize dynamics. It also makes it clear why it's more effective when dispersal is active, since it then acts as a direct density-dependent mortality rate, and as McCann (2011) and others have shown, direct density-dependent negative feedbacks will generally be strongly stabilizing. This would be a good place to highlight what the drainage mechanism is actually doing at the landscape scale.

McCann, K.S. (2011). Food webs. Princeton University Press.

Line 149: "dispersal loss": I agree with the author's decision in their response to talk about dispersal loss rather than just dispersal mortality; their point that dispersal loss entails both dispersal mortality and energetic losses from movement is one that I had not thought about when writing my prior review. I would suggest that the authors clarify here what they mean by dispersal loss, though, and emphasize that it is this non-conservative effect that is causing the drainage effect.

Figure 4: I have found the explanation for this figure to be a bit hard to follow. When I first went through the figure, I thought that the curves represented average alpha diversity among patches for a given simulation, rather than local alpha diversity. It would be useful to clarify that the y-axis represents alpha diversity observed in a single patch (rather than average alpha diversity across patches in the simulation), and that nutrient supply is local nutrient supply at a single patch (rather than averages across landscapes). I would also suggest changing the points so that the

size or color of the points indicates the total number of patches across simulations with that combination of nutrient supply and alpha-diversity. Currently the points convey very little information, and because they involve over-plotting a large number of individual points, take a lot of time to render (I found that the supplemental file was very slow to render because of this).

Line 387-396: Numerical simulations: in the response to reviewers, the 100,000 time step simulation period was noted to correspond to ~11 years; this information should be included here. It seems unlikely that any transient dynamics may have equilibrated in this complex a food web. Are the results in figure 4 robust to a longer simulation period?

Line 405: Since both sides of all three equations have to be in the same units, I do not think it is necessary to include units next to all of the model terms.

Signed,

Eric Pedersen

Reviewer #3:

Remarks to the Author:

I think that the revised manuscript has been improved. However, I still have two main concerns.

(1)

As the authors indicate in their response to my comment, the drainage effect demonstrated in their food chain models (Fig. 2 and Fig. 3) is not entirely novel. This is not clear in the revised manuscript, which might mislead the readers. I suggest the followings:

(1.1) Lines 101 and 230: Remove "novel"s.

(1.2) Lines 125-128: State here explicitly that this hypothesis is derived from the existing theories on the effect of consumer dispersal on preventing extinctions.

(2)

In my comments on the previous manuscript, I suggested that the true novelty of this manuscript might be that it shows the synergistic effects of rescue and drainage effects in a complex food web meta-community framework (Fig. 4). The revised manuscript has incorporated this point. However, unclarity remains. I suggest three points.

(2.1) Line 236: The authors maintain that their study focuses on the biodiversity of local patches. This statement seems to contradict with Fig. 3a, which focuses on the number of populations in the whole landscape.

(2.2) Line 239-241: I am afraid that the explanation on Fig. 4 might be insufficient. Consider rephrasing from "purple line ... the violin plot" to "the local diversity on the purple line at the average nutrient supply level of orange landscapes indicated by the orange diamond in the violin plot."

(2.3) To fully establish their findings, it would be highly valuable if the authors could carry out a sensitivity analysis regarding dispersal modes on the results of Figure 4.

REVIEWER COMMENTS

Reviewer #1 (Remarks to the Author):

I appreciate all the efforts that the authors have made to improve their work, the manuscript looks much better and stronger now.

We are very thankful for the insightful discussion that has helped to strengthen our manuscript substantially.

Reviewer #2 (Remarks to the Author):

This is my second time reviewing this paper. The authors have thoughtfully responded to most of my and the other reviewer's comments on the paper. I do still have some issues with how nutrient heterogeneity is being included in the model, as well as with how well the drainage effect has been connected to prior research on stabilizing mechanisms in food webs.

Our manuscript has profited substantially from the astute observations and insightful comments made by the reviewers. We highly appreciate this discussion.

My primary concern is still regarding the basis for how heterogeneous nutrient supply rates were included in the modelling framework. In the author's response to my initial comments, they noted that they focused on the drainage effect from a patch-perspective, which I take to mean that they focused on the effect of the drainage effect on the eutrophic patch. I agree that this is a reasonable framing for figure 3b. However, the focus of figure 3a) and possibly figure 4) are on the coexistence of multiple species across a landscape, which is very much not a patch-perspective, and will be heavily affected by average nutrient supplies to the entire landscape. Figure S6 seems to imply that fluctuation magnitudes are not that sensitive to heterogeneity when total input rates are kept constant. This should be something noted in the main text, as the stabilizing effects of reducing total energy inputs into a system are a well-known phenomenon.

We appreciate this input and apologize for our previous unclarity. We followed this suggestion concerning Fig. 4 (see below) and clarified that the main focus of our work is the drainage effect on local diversity (and not regional diversity). Additionally, we elaborated on the effect of landscape-average nutrient supply and refer to the corresponding supplementary Figure (now Fig. S7) on L251-268. Also see response to Reviewer 3, point 2.1.

Specific comments:

Line 24: "'drainage effect' stabilizes biodiversity by more uniform distributions": this phrasing is unclear. I would rephrase this to clarify what the drainage effect is. Something along the lines of "'drainage effects' due to high dispersal mortality from abundant

patches stabilize biodiversity dynamics of eutrophic patches. This effect is magnified in heterogeneous landscapes.”

We agree that this statement has been unclear. Unfortunately, we could not follow the suggestion above as this would have been above the word limit of the abstract. We have revised this sentence to: “Second, the “drainage effect” stabilizes biodiversity by preventing overshooting of population densities on eutrophic patches.” (L23-25).

Line 129: It would be useful to make the direct connection here that emigration from these patches in the single-patch model is effectively acting as an additional source of mortality; since the emigrants are being lost to the system, emigration is indistinguishable from mortality. Since this will reduce the average growth rate of all species, it should naturally act to stabilize dynamics. It also makes it clear why it’s more effective when dispersal is active, since it then acts as a direct density-dependent mortality rate, and as McCann (2011) and others have shown, direct density-dependent negative feedbacks will generally be strongly stabilizing. This would be a good place to highlight what the drainage mechanism is actually doing at the landscape scale.

McCann, K.S. (2011). Food webs. Princeton University Press.

We thank the reviewer for this input and made the connection to mortality rates for dispersal (L124-134; including the suggested reference) and in regard to nutrient supply heterogeneity (L158 -162).

Line 149: “dispersal loss”: I agree with the author’s decision in their response to talk about dispersal loss rather than just dispersal mortality; their point that dispersal loss entails both dispersal mortality and energetic losses from movement is one that I had not thought about when writing my prior review. I would suggest that the authors clarify here what they mean by dispersal loss, though, and emphasize that it is this non-conservative effect that is causing the drainage effect.

We now explicitly state the two aspects of dispersal loss (i.e. energetic losses of movement and dispersal mortality (e.g. road kills)). In addition, and by adapting the suggestion above, we relate the drainage effect to indirect mortality rates.

Figure 4: I have found the explanation for this figure to be a bit hard to follow. When I first went through the figure, I thought that the curves represented average alpha diversity among patches for a given simulation, rather than local alpha diversity. It would be useful to clarify that the y-axis represents alpha diversity observed in a single patch (rather than average alpha diversity across patches in the simulation), and that nutrient supply is local nutrient supply at a single patch (rather than averages across landscapes). I would also suggest changing the points so that the size or color of the points indicates

the total number of patches across simulations with that combination of nutrient supply and alpha-diversity. Currently the points convey very little information, and because they involve over-plotting a large number of individual points, take a lot of time to render (I found that the supplemental file was very slow to render because of this).

We followed these suggestions that improved the figure substantially.

Line 387-396: Numerical simulations: in the response to reviewers, the 100,000 time step simulation period was noted to correspond to ~11 years; this information should be included here. It seems unlikely that any transient dynamics may have equilibrated in this complex a food web. Are the results in figure 4 robust to a longer simulation period?

We have included this information there (L398-412). In most cases, transient dynamics equilibrate after the first few thousand time steps (although some continue to oscillate, e.g. simulations Fig 3, amplitude > 0). For all simulations we checked (although not systematically) the averaged net-growth of each species on each patch the time steps between 80'000 and 90'000, and between 90'000 and 100'000. In the vast majority, this was 0 or very small (below 10-20). These 11 years should only be compared with caution to natural time scales of population dynamics. Although the metabolic rate was derived empirically, our model does not include non-trophic activities such as mating or sleeping that are time consuming for organisms.

Line 405: Since both sides of all three equations have to be in the same units, I do not think it is necessary to include units next to all of the model terms.

done

Signed,

Eric Pedersen

Reviewer #3 (Remarks to the Author):

I think that the revised manuscript has been improved. However, I still have two main concerns.

(1) As the authors indicate in their response to my comment, the drainage effect demonstrated in their food chain models (Fig. 2 and Fig. 3) is not entirely novel. This is not clear in the revised manuscript, which might mislead the readers. I suggest the followings:

(1.1) Lines 101 and 230: Remove "novel"s.

done

(1.2) Lines 125-128: State here explicitly that this hypothesis is derived from the existing theories on the effect of consumer dispersal on preventing extinctions.

Following this suggestion by Reviewer 2, we stated this explicitly on L124 -128 and implicitly by more clearly relating the drainage effect to density-dependent negative feedbacks on L130-132 and L158-162

(2) In my comments on the previous manuscript, I suggested that the true novelty of this manuscript might be that it shows the synergistic effects of rescue and drainage effects in a complex food web meta-community framework (Fig. 4). The revised manuscript has incorporated this point. However, unclarity remains. I suggest three points.

(2.1) Line 236: The authors maintain that their study focuses on the biodiversity of local patches. This statement seems to contradict with Fig. 3a, which focuses on the number of populations in the whole landscape.

This is an astute observation. We have changed the plot in Fig. 3a to show the average number of populations on the two patches (i.e., the average biodiversity on local patches) rather than the number of populations on both patches (see revised Fig. 3a).

(2.2) Line 239-241: I am afraid that the explanation on Fig. 4 might be insufficient. Consider rephrasing from "purple line ... the violin plot" to "the local diversity on the purple line at the average nutrient supply level of orange landscapes indicated by the orange diamond in the violin plot."

We have followed this suggestion.

(2.3) To fully establish their findings, it would be highly valuable if the authors could carry out a sensitivity analysis regarding dispersal modes on the results of Figure 4.

We performed additional simulations in the complex landscapes (as in Fig. 4) using a non-adaptive dispersal model. This figure is now included in the Supplement (Fig. S5) and the results are virtually indistinguishable from the results shown in Fig. 4 of the main manuscript.

Reviewers' Comments:

Reviewer #2:

Remarks to the Author:

I am satisfied with how the authors addressed my and other reviewers' comments, and I commend the effort and thought they put into this revision. I have no further comments, besides noting a couple minor typos in the text, on line 126 ("highlighted") and 159 ("arraises").

Reviewer #3:

Remarks to the Author:

This second revision has been improved well. I would appreciate the effort of the authors to address all the points made by the reviewers.

REVIEWERS' COMMENTS

Reviewer #2 (Remarks to the Author):

I am satisfied with how the authors addressed my and other reviewers' comments, and I commend the effort and thought they put into this revision. I have no further comments, besides noting a couple minor typos in the text, on line 126 ("highlighted") and 159 ("arraises").

We appreciate this feedback and thank the reviewer for their effort and thought too. We corrected the two typos.

Reviewer #3 (Remarks to the Author):

This second revision has been improved well. I would appreciate the effort of the authors to address all the points made by the reviewers.

We thank the reviewer for this feedback.